

# Pysteps: an open-source Python library for probabilistic precipitation nowcasting (v1.0)

Seppo Pulkkinen[1,2], Daniele Nerini[3,4], Andrés A. Pérez Hortal[5], Carlos Velasco-Forero[6], Alan Seed[6], Urs Germann[3], and Loris Foresti[3]

[1]Colorado State University, Fort Collins, United States
[2]Finnish Meteorological Institute, Helsinki, Finland
[3]Federal Office of Meteorology and Climatology MeteoSwiss, Locarno-Monti, Switzerland
[4]Institute for Atmospheric and Climate Science, ETH Zurich, Switzerland
[5]Department of Atmospheric and Oceanic Sciences, McGill University, Montreal, Canada
[6]Bureau of Meteorology, Melbourne, Australia

**Correspondence:** S. Pulkkinen (seppo.pulkkinen@fmi.fi)

**Abstract.** Pysteps is an open-source and community-driven Python library for probabilistic precipitation nowcasting– that is to say, very-short range forecasting (0-6 h). The aim of pysteps is to serve two different needs. The first is to provide a modular and well-documented framework for researchers interested in developing new methods for nowcasting and stochastic space-time simulation of precipitation. The second aim is to offer a highly configurable and easily accessible platform for practitioners ranging from weather forecasters to hydrologists. In this sense, pysteps has the potential to become an important component for integrated early warning systems for severe weather.

The pysteps library supports standard input/output file formats and implements several optical flow methods as well as advanced stochastic generators to produce ensemble nowcasts. In addition, it includes tools for visualizing and post-processing the nowcasts and methods for deterministic, probabilistic, and neighbourhood forecast verification. The pysteps library is described and its potential is demonstrated using radar composite images from Finland, Switzerland, United States, and Australia. Finally, scientific experiments are carried out to help the reader to understand the pysteps framework and sensitivity to model parameters.

## 1 Introduction

As defined by the world meteorological organization, nowcasting encompasses a description of the current state of the atmosphere along with forecasts obtained by extrapolation for up to 6 hours ahead. As such, it represents an essential tool to forecast severe weather, as for example, heavy precipitation and intense thunderstorms.

Excessive rainfall can act as a trigger for water-related hazards (Alfieri et al., 2012), and this is particularly true in an increasingly urbanized territory or in presence of steep topography. When vulnerable objects are exposed to such hazards, risks can manifest as property damage, economic losses and loss of lives.

Examples of scenarios where reliable precipitation nowcasts are needed include the decision to interrupt a train line exposed to debris flow, evacuation of buildings in flood-prone areas, optimization of airport operations in severe weather and regulation



of sewage systems during storm events. All such scenarios can benefit from the development of effective real-time nowcasting systems that take into account the predictability of precipitation and related hazards at a high spatial and temporal resolution so that risk be mitigated.

## 1.1 From deterministic to probabilistic nowcasting

5 Weather radars are ideally suited for providing the input data for extrapolation-based precipitation nowcasting at high resolution, namely spatial scales under 2 km and time ranges between 5 minutes and 3 hours (Berne et al., 2004). Despite recent advances in numerical weather prediction (NWP, e.g. Sun et al., 2014), extrapolation-based nowcasting remains the primary approach at such space and time scales. In fact, many users and applications require a level of forecast accuracy at the convective scale that is currently difficult to reach with NWP (e.g. Simonin et al., 2017).

10 Precipitation exhibits variability over a wide range of space-time scales (e.g. Lovejoy and Schertzer, 2013) which, in combination with the chaotic nature of the atmosphere (e.g. Lorenz, 1996), limits our ability to predict its evolution in a deterministic manner. The NWP community recognized this challenge in the early 1990s and tackled the problem by producing an ensemble of NWP forecasts by perturbing the initial conditions (e.g. Toth and Kalnay, 1997). Those perturbations grow exponentially and lead to an ensemble of solutions that reflect forecast uncertainties. The information contained in the ensemble can then be 15 used to derive probabilistic forecasts.

Following the NWP developments, the nowcasting community also acknowledged the importance of estimating predictive uncertainty (e.g. Seed, 2003; Germann and Zawadzki, 2004; Bowler et al., 2006). A common approach is based on stochastic simulation, in which correlated noise fields are used to perturb a deterministic nowcast (e.g. Bowler et al., 2006; Berenguer et al., 2011; Liguori and Rico-Ramirez, 2014; Foresti et al., 2016). Substantial research efforts have been made to make the 20 perturbation fields as realistic as possible, and consistent with the nowcast uncertainty (e.g. Seed et al., 2013; Nerini et al., 2017). For a review of the history of nowcasting since the 1950s, and its evolution to the probabilistic framework, we refer the reader to Pierce et al. (2012).

## 1.2 The pysteps open-source initiative

Similarly to other research fields, the nowcasting community has invested a significant amount of time to re-implement from 25 scratch routines and algorithms that have been around for decades, for example, optical flow and advection schemes. Part of this problem is due to the unavailability of software, which is often proprietary or too poorly documented to be understood, trusted, and used.

Recognizing that nowcasting methods and related applications can be further developed and distributed by promoting universal access to existing knowledge, a Python-based software package, called *pysteps*, is being developed as a community-driven 30 effort. This effort fits well into the weather radar community with emergence of open data and an increasing number of open-source software projects (Heistermann et al., 2015), for instance, in radar data processing (Heistermann et al., 2013; Helmus and Collis, 2016). More recently, community-based initiatives dedicated to nowcasting have emerged, as for example com-





SWIRLS by the Hong Kong Observatory, IMPROVER by the UK MetOffice or rainymotion at the University of Potsdam (see Table 1).

The objective of pysteps is twofold. First, it aims at providing a well-documented and modular framework for development of new nowcasting methods. In this sense, pysteps promotes the adoption of open-science practices, as the lack of common standards, transparency, code availability and well-documented workflows in computational disciplines can lead to non-reproducible results, hence questioning their scientific value (Hutton et al., 2016). Second, pysteps aims at providing an easily accessible software package for practitioners ranging from weather forecasters to hydrologists.

### 1.3 Outline of the paper

The paper is structured as follows. The theoretical framework for precipitation nowcasting and using stochastic perturbations to characterize the uncertainty is formulated in Section 2. The general architecture of the pysteps library is presented in Section 3. A comprehensive verification of pysteps nowcasts is given in Section 4. Various experiments to understand the sensitivity of pysteps to the model parameters and define the default configuration are done in Section 5. The limits of pysteps are tested in Section 6 using a tropical cyclone and severe convection case in Australia. Section 7 concludes the paper and lists potential future applications of pysteps. Finally, code listings demonstrating the use of pysteps are given in Appendix A.

## 2 Formulation of precipitation nowcasting

This section introduces the basic concepts and components of probabilistic nowcasting models based on the Lagrangian persistence of radar precipitation fields and describes how these are currently implemented in pysteps.

### 2.1 Lagrangian persistence and optical flow

In its simplest form, extrapolation-based precipitation nowcasting assumes that over the time frame of a few hours the evolution of precipitation can be captured by moving the radar echoes along a stationary motion field without changes in intensity. In the literature, this is known as *Lagrangian persistence* (e.g. Germann and Zawadzki, 2002).

Denoting a precipitation parcel by $R$ and its displacement vector by $\boldsymbol{\alpha}(\tau)$, the conservation equation for an incompressible flow is written as

$$R(\boldsymbol{x}_0; t + \tau) = R(\boldsymbol{x}_0 - \boldsymbol{\alpha}(\tau); t), \tag{1}$$

or equivalently in differential form as

$$\frac{dR}{dt} = \frac{\partial R}{\partial t} + u\frac{\partial R}{\partial x} + v\frac{\partial R}{\partial y}, \quad u = \frac{dx}{dt}, \quad v = \frac{dy}{dt}, \tag{2}$$

where $dR/dt = 0$, and $u$ and $v$ are the x- and y-components of the motion field. In the so-called optical flow methods, $u$ and $v$ are estimated for a given location by solving equation (2) numerically based on a sequence of precipitation intensity fields.





Typically, a constraint on the spatial continuity of nearby $u$ and $v$ is imposed to guarantee a unique solution. Once the motion field is known, the radar echoes are extrapolated by means of an advection scheme.

Three methods are currently implemented in pysteps for motion field estimation: a local Lucas-Kanade method (Lucas and Kanade, 1981; Bouguet, 2001), a global variational echo tracking approach (Laroche and Zawadzki, 1994; Germann and
Zawadzki, 2002), and a spectral approach (DARTS, Ruzanski et al., 2011). The currently implemented advection method is the backward-in-time semi-Lagrangian scheme described in Germann and Zawadzki (2002), which is robust against numerical diffusion.

## 2.2 Sources of uncertainty

The predictability of the atmosphere is intrinsically limited by the fact that its state cannot be observed with absolute precision
nor expressed without approximations in its governing laws (Lorenz, 1996). In the case of radar-based precipitation nowcasting, predictive uncertainty originates from errors in the estimation of the *current state* of the rainfall and motion fields (initial state errors), and limitations of Lagrangian persistence as a model to predict the *evolution* of the rainfall and motion fields (model errors).

The main contribution to model errors in the Lagrangian approach stems from the evolution of precipitation in terms of
initiation, growth, decay and termination processes that violate the steady-state assumption. Other sources of model uncertainty include the assumption of stationarity of the motion field, inaccuracies due to the practical implementation of the method, as the discretization in time, space and reflectivity, and numerical diffusion of the advection scheme (Germann et al., 2006b).

Currently, pysteps focuses on the representation of the model errors, whereas incorporation of the initial state errors in the nowcasting is left for future work.

## 2.3 Data transformation

The statistics of intermittent precipitation rates are non-Gaussian and display a typical asymmetric distribution that is bounded at zero. Such properties restrict the usage of well-established stochastic models that assume Gaussianity. A common workaround is to introduce a suitable data transformation to approximate a normal distribution (e.g. Erdin et al., 2012).

Currently, pysteps assumes a log-normal distribution of rain rates by applying the logarithmic transformation

$$R \to \begin{cases} 10\log_{10} R, & \text{if } R \geq 0.1 \text{ mm h}^{-1} \\ -15, & \text{otherwise} \end{cases} \qquad (3)$$

that corresponds to logarithmic radar rain rates (units of dBR). The value of -15 dBR is equivalent to assigning a rain rate of approximately 0.03 mm h$^{-1}$ to the zeros. Hereafter, $R$ refers to the transformed rain rates, unless otherwise stated.

Using the logarithmic transformation is motivated by the fact that rain rates are approximately log-normally distributed (Crane, 1990). This has two main advantages. First, it simplifies the estimation of distribution parameters, particularly with
limited sample size and in presence of measurement noise (Harris et al., 1997). Second, the decomposition of log-transformed



rainfall fields defines a multiplicative cascade, where multiplications are replaced with summations in the transformed space (Seed, 2003).

## 2.4  A cascade of spatial scales

It has been shown that the lifetime of precipitation relates to its spatial scale (e.g., Venugopal et al., 1999; Seed, 2003; Germann
et al., 2006b), often denominated as dynamic scaling. Recognizing this fundamental property, Seed (2003) introduced the S-PROG model, which laid the foundation for the development of STEPS (Bowler et al., 2006; Seed et al., 2013). The key idea is to decompose the precipitation field into a multiplicative cascade, where the cascade levels represent different spatial scales, and treat them separately in the nowcasting model.

In STEPS, the scale decomposition is done by applying a Fast Fourier Transform (FFT) to the input precipitation field. This
is motivated by the fact that for a grid of size $L \times L$ pixels, the radial Fourier wavenumbers $|\boldsymbol{k}| = \sqrt{k_x^2 + k_y^2}$ are related to spatial scales via

$$
\underbrace{|\boldsymbol{k}|}_{\substack{\text{radial} \\ \text{wavenumber} \\ \text{(pixels)}}} \quad \rightarrow \quad \underbrace{\frac{L}{|\boldsymbol{k}|}}_{\substack{\text{wavelength} \\ \text{(pixels)}}} \quad \rightarrow \quad \underbrace{\frac{L\Delta x}{|\boldsymbol{k}|}}_{\substack{\text{wavelength} \\ \text{(km)}}} \quad \rightarrow \quad \underbrace{\frac{L\Delta x}{2|\boldsymbol{k}|}}_{\substack{\text{scale} \\ \text{(km)}}} \;, \tag{4}
$$

where $\Delta x$ denotes the grid resolution (km). Thus, the spatial scale is half the wavelength. Alternative approaches to perform a scale decomposition include the Discrete-Cosine-Transform (Germann and Zawadzki, 2002; Surcel et al., 2014) or wavelets
(Turner et al., 2004; Scovell, 2018).

In the current implementation of pysteps, we adopt the approach of Pulkkinen et al. (2018), where Gaussian weight functions are used for separating the Fourier spectrum into a set of radial bands. An example of the weight functions for the domain covered by the Finnish Meteorological Institute (FMI) radars is shown in Fig. 1. After the FFT and Gaussian filtering, each frequency band is transformed back to the spatial domain, which results in a cascade with $n$ levels each representing a different
scale (see an example in Fig. 2).

## 2.5  Temporal evolution

In nowcasting, the typical approach to model the temporal evolution of precipitation fields employs an auto-regressive (AR) process that combines the deterministic component from Lagrangian persistence with a stochastic innovation term, also referred to as noise or perturbation term. For instance, S-PROG and STEPS use a second-order AR(2) process with two parameters.
Separate AR(2) processes are applied to each cascade level to account for the dynamic scaling of precipitation. The combination of the auto-regressive model in time and the cascade model in space allows one to control the temporal evolution and correlation structure of precipitation.





Currently, a more general AR(p) model has been implemented in pysteps. For each cascade level $j$, the recursion formula is given by

$$R_j(x,y,t) = \sum_{k=1}^{p} \phi_{j,k} R_j(x,y,t-k\Delta t) + \\ \phi_{j,0} \varepsilon_j(x,y,t). \tag{5}$$

The first term corresponds to the deterministic predictable component at cascade level $j$ (i.e. Lagrangian persistence). The

second term is a stochastic term that represents the unpredictable component at the same cascade level $j$, that is, mainly initiation, growth and decay of precipitation. The symbol $\Delta t$ denotes the time difference between consecutive precipitation fields $R_j$ that are normalized to zero mean and unit variance.

The parameters $\phi_{j,k}$ in the above model are estimated from time-lagged auto-correlation coefficients $\rho_{j,k}$ for $k = 1, 2, \ldots, p$ using the Yule-Walker equations (Hamilton, 1994). For $p = 2$, the correlation coefficients can be adjusted to ensure that the

resulting AR(p) process is stationary and non-periodic (Box et al., 2013). Finally, the parameters $\phi_{j,0}$ are chosen as

$$\phi_{j,0} = \sqrt{1 - \sum_{k=1}^{p} \rho_{j,k} \phi_{j,k}}. \tag{6}$$

Given that the variance of the noise fields $\varepsilon_j$ is one, this choice guarantees that the AR(p) process is normalized to unit variance (Hamilton, 1994).

The theoretical auto-correlation function (ACF) of the AR(2) process can be computed recursively from the model parame-

ters and auto-correlation coefficients (Chatfield, 2003) according to

$$\rho_j(t) = \phi_{j,1} \rho_j(t - \Delta t) + \phi_{j,2} \rho_j(t - 2\Delta t). \tag{7}$$

The empirical ACF can be derived by computing the correlation coefficients between the extrapolation nowcasts and the observations.

For an exponentially decaying ACF, the precipitation lifetime is defined as the time when the ACF, theoretical or empirical,

falls below the value $1/e \approx 0.37$, where $e$ is the Euler number. Alternatively, one can estimate the lifetime by integrating the ACF according to

$$T = \int_0^\infty \rho(\tau) d\tau. \tag{8}$$

It is not common to employ an AR(p) process with $p > 2$ for several reasons. First, it is not trivial to guarantee the stationarity and non-periodicity of the process. Second, when estimated in Lagrangian frame, the higher-order auto-correlation coefficients

are affected by the uncertainty of the motion field. This occurs especially at small spatial scales as it is difficult to properly track convective cells over several time steps. Third, a low-order AR process is generally sufficient to model the loss of predictability in the nowcasting range; departures are usually observed only after $\approx 2$ hours (Atencia and Zawadzki, 2014).





## 2.6 Stochastic perturbations of precipitation intensities

The perturbation field $\varepsilon$ in equation (5) is typically generated as a correlated Gaussian random field using FFT filtering (e.g. Pegram and Clothier, 2001; Bowler et al., 2006). The process consists of three steps:

1. generate a Gaussian white noise field,

2. apply the FFT and a Fourier filter to the above to generate a random field having the desired correlation structure,

3. apply the inverse FFT to transform the noise field back to the spatial domain.

This technique is also known as power-law filtering of white noise or fractional integration (Schertzer and Lovejoy, 1987).

At present, three methods for filtering white noise fields have been implemented in pysteps. In the absence of a model that predicts the evolution of the spatial correlation structure, one assumes that the correlation structure remains constant through
the nowcast. An example is provided in Fig. 3.

In the *parametric method* introduced by Pegram and Clothier (2001), the filtered noise field $\varepsilon$ is obtained from the white noise field $\varepsilon_w$ as

$$\varepsilon(x,y) = \mathcal{F}^{-1}\{f(|\boldsymbol{k}|)\mathcal{F}\{\varepsilon_w\}(k_x,k_y)\}, \tag{9}$$

where $\mathcal{F}$ denotes the Fourier transform and the function $f$ defines the slope of the radially averaged power spectrum (RAPS).
Our implementation follows the approach by Seed (2003), which uses a piece-wise linear function with two spectral slopes $(\beta_1,\beta_2)$ and one breaking point. The main limitation of such model relates to the assumption of an isotropic power law scaling relationship, meaning that anisotropic structures such as rainfall bands cannot be represented.

In the *non-parametric method* (Seed et al., 2013), the Fourier filter is obtained directly from the power spectrum of the observed precipitation field $R$ such that

$$\varepsilon(x,y) = \mathcal{F}^{-1}\{|\mathcal{F}\{R\}(k_x,k_y)|\mathcal{F}\{\varepsilon_w\}(k_x,k_y)\}. \tag{10}$$

Differently to the parametric method, the non-parametric approach allows generating perturbation fields with anisotropic structures. On the other hand, the approach requires a larger sample size and is sensitive to the quality of the input data, e.g. the presence of residual clutter in the radar image. In addition, both techniques assume spatial stationarity of the covariance structure of the field.
The third method is an extension of the non-parametric approach, where the noise field is generated *locally* to account for spatial inhomogeneities in the covariance structure of the rainfall field. The technique is based on the short-space Fourier transform (SSFT) described in Nerini et al. (2017). Essentially, the non-parametric approach in (10) is localized in $(x,y)$ by

$$\varepsilon(x,y) = \mathcal{F}^{-1}\{|\mathcal{F}\{Rw_h(n_1,n_2)\}(k_x,k_y)|\mathcal{F}\{\varepsilon_w\}(k_x,k_y)\}, \tag{11}$$

where $w_h(n_1,n_2) = w_h(n_1)w_h(n_2)$ is the outer product of two Hanning windows of size $n_1$ and $n_2$ centred in $(x,y)$.





## 2.7 Stochastic perturbations of the motion field

A second source of uncertainty in Lagrangian persistence nowcasting stems from temporal evolution of the motion field (Germann et al., 2006b). This can be accounted for by adding stochastic perturbations. In the current implementation of pysteps, this is done by applying the method of Bowler et al. (2006).

For simplicity, the perturbation field is assumed to be spatially constant for each ensemble member, but the magnitude of the perturbations increases with respect to lead time. For a given initial advection field $\boldsymbol{w}_0$ and lead time $t$, the perturbed velocities are given by

$$
\begin{aligned}
\boldsymbol{w}_p(x,y) = {}& \boldsymbol{w}_0(x,y) + \alpha_{\mathrm{par}}(t)\varepsilon_{\mathrm{par}}(x,y)\hat{\boldsymbol{w}}_{\mathrm{par}} + \\
& \alpha_{\mathrm{perp}}(t)\varepsilon_{\mathrm{perp}}(x,y)\hat{\boldsymbol{w}}_{\mathrm{perp}},
\end{aligned}
\tag{12}
$$

where $\hat{\boldsymbol{w}}_{\mathrm{par}}$ and $\hat{\boldsymbol{w}}_{\mathrm{perp}}$ denote the components parallel and perpendicular to the initial advection field $\boldsymbol{w}_0$, respectively. The
random variables $\varepsilon_{\mathrm{par}}$ and $\varepsilon_{\mathrm{perp}}$ are sampled from the Laplace distribution with zero mean and unit variance. Scaling of the perturbations is done according to

$$
\alpha_{\mathrm{par}}(t) = a_{\mathrm{par}}t^{b_{\mathrm{par}}} + c_{\mathrm{par}}
\tag{13}
$$

$$
\alpha_{\mathrm{perp}}(t) = a_{\mathrm{perp}}t^{b_{\mathrm{perp}}} + c_{\mathrm{perp}},
\tag{14}
$$

where the parameters are climatologically fitted by using a large sample of advection fields.

## 2.8 Post-processing of nowcasts

To ensure that the forecast fields have the same statistical properties with the observed ones, post-processing is typically done at the very end of the chain. This is necessary because intermediate steps may introduce discrepancies. One major source of such discrepancies is related to the difficulty to model the intermittency of precipitation. Typically, the basic statistical properties such as wet-area ratio, mean, variance and the marginal distribution of precipitation intensities are assumed to remain invariant
through the nowcast.

In the present implementation of pysteps the post-processing involves two different types of methods: 1) masking and 2) matching the statistics of the forecast fields with the most recently observed ones. Methods of type 1) are used to avoid generation of stochastic cells into areas that are too distant from existing precipitation. Methods of type 2) can be applied unconditionally or only to pixels within the mask.

Three different masking methods have been implemented. In the first method, the mask is obtained from pixels exceeding an intensity threshold in the observed precipitation field, and the mask is kept constant in Lagrangian coordinates for the whole forecast. In the second method adapted from Seed (2003), the mask is obtained by using the S-PROG (i.e. the unperturbed STEPS) nowcast. In the third method, a lead-time-dependent precipitation mask is applied. The mask is defined by the pixels exceeding a given intensity threshold in the observed precipitation field and then progressively relaxed to allow the stochastic
evolution of the wet area.





Two methods have been implemented for matching the statistics of forecast fields with the observed ones. In the first method, that is used together with the S-PROG mask, the conditional mean of the masked forecast field is adjusted to match the conditional mean of the observed field (excluding intensities below the threshold). Alternatively, the cumulative distribution function (CDF) of the forecast field can be mapped to the observed one. This is defined as

$$R'(x,y) = F_{\mathrm{obs}}^{-1}(F(R(x,y))), \tag{15}$$

where $F_{\mathrm{obs}}$ and $F$ denote the CDFs of the observed and the input forecast field $R$, respectively. This approach is known as probability matching or quantile mapping.

## 3 The pysteps library

### 3.1 Key features and development model

The implementation language of pysteps is Python (http://python.org). As a high-level language with an extensive built-in standard library and a large number of external libraries available, it is ideally suited for open-source software development. Python distributions, such as Anaconda, providing the necessary software to run pysteps are available for all major platforms. Python also provides interfaces for compiled languages such as C/C++ and Fortran, allowing to improve performance in time-critical modules. In addition, Python-based tools, like the IPython shell (Pérez and Granger, 2007) or the Jupyter notebooks (https://jupyter.org), allow an interactive use of pysteps for research and demonstration purposes.

The pysteps library is extensively documented. The documentation describes in detail the different modules and the application programming interfaces (API). The modules are documented by using the docstring concept of Python. This is implemented using Read the docs (https://readthedocs.org/) and Sphinx (http://www.sphinx-doc.org/en/master) to automatically compile and update an online version of the documentation, available at https://pysteps.readthedocs.io. In addition, tutorials for performing various tasks with pysteps are included as example scripts.

Pysteps development is done by using git, a distributed version control system. The source code of pysteps is hosted in GitHub (https://pysteps.github.io). In addition to code hosting, the features of GitHub include development in multiple branches, issue tracking and wiki pages. Developers outside the core team may fork the main repository and integrate the proposed changes via pull requests, which allows community-driven development. Continuous integration and testing is done by using the Travis CI framework (https://travis-ci.com/pySTEPS/pysteps).

Pysteps is published under the 3-Clause BSD License. It allows copying, redistribution and modification of the software as long as the modification are tracked and the source code is made available under the the same license. The permissive license model makes the software easily accessible to potential users, even allowing use for commercial purposes.

### 3.2 External dependencies

Pysteps relies on several external libraries that are listed in Table 2. It is built on top of numpy, scipy and matplotlib, that together provide a MATLAB-like computing environment in Python. These libraries provide data structures and wrappers for



low-level BLAS and LAPACK libraries for high-performance matrix and array operations, image processing methods and also high-level functionality for data visualization. The numpy array is the basic data structure used in pysteps.

Support for netCDF (the default file format), HDF5, and various image file formats is implemented via the netCDF4, h5py, and PIL libraries. Plotting precipitation data with basemaps has been implemented via mpl_toolkits.basemap and cartopy packages. The Lucas-Kanade optical flow algorithm used in pysteps is implemented in the OpenCV library and accessed via a Python interface. Parallelized computation of nowcast ensembles is done by using dask that provides a platform-independent backend for low-level methods.

### 3.3 Key design principles

The aim of pysteps is to be a modular software library, where all the main components are interchangeable, thus making it an ideal research platform for developing and testing new methods. Pysteps is currently divided into 11 modules that perform different tasks. The modules and their descriptions are listed in Table 3.

The modularity is implemented via interface-based design. To this end, each module implements one sub-task and an interface method for retrieving the desired method for this task. All mutually interchangeable methods implement the same interface. Another key principle is that whenever possible, the data is stored into n-dimensional arrays, which allows an efficient and compact representation.

The above design principles are demonstrated in the following example. A precipitation nowcast by using STEPS can be generated by

```
>>> nowcast_method = nowcasts.get_method("steps")
>>> nowcast = nowcast_method(R, V, num_timesteps)
```

where the required inputs are

| | |
|---|---|
| R | array of shape (t,m,n) containing a time series of t observed precipitation intensity fields with shape (m,n) |
| V | a previously computed array of shape (2,m,n) containing the x- and y-components of the advection field |
| num_timesteps | number of time steps to forecast |

Additional parameters can be specified by using keyword arguments. The output of stochastic nowcasting methods is a four-dimensional array of shape (num_ensemble_members, num_timesteps, height, width). For deterministic nowcasts, the first dimension is dropped.

### 3.4 Data structures

In addition to being modular, pysteps implements object-oriented features. However, instead of using customized classes, we use dictionaries and functions that operate on the dictionaries similarly to class member functions. This design decision is motivated by the principle of using the core Python datatypes rather than implementing customized classes. The flat design of pysteps should facilitate user interaction and embedding of individual modules and functions in other software. In this way, pysteps is similar to wradlib (Heistermann et al., 2013).





To demonstrate the above design, the following example shows how to construct a Gaussian bandpass filter for 8 cascade levels using the `filter_gaussian` function implemented in the `cascade.bandpass_filters` module:

```
>>> filter = filter_gaussian(R.shape, 8)
```

The output is a dictionary with three elements: 1) one-dimensional weights corresponding to the radial wavenumbers, 2) a two-dimensional weight field for the FFT of the input image, and 3) a list of central frequencies for each weight function (see

Fig. 1). The resulting filter object can then be passed to `decomposition_fft` as follows:

```
>>> decomp = decomposition_fft(R, filter)
```

The decomposition is applied to a two-dimensional precipitation field R, and the output is a again a dictionary with three-elements: 1) a three-dimensional array containing the 8 cascade levels having the same dimension as R, 2) mean precipitation values of each cascade level and 3) standard deviations for each level. More detailed examples of pysteps usage are provided in Appendix A.

**3.5  Workflow**

Figure 4 gives a schematic view of the workflow for generating precipitation nowcasts using pysteps. The first step is reading the radar composites by using the methods implemented in the `io` module. This is followed by determination of the motion field using the methods implemented in the `motion` module.

The radar composites and the motion field are supplied as inputs to a user-chosen nowcasting method implemented in the

`nowcasts` module. For the Lagrangian persistence method implemented in `nowcasts.extrapolation`, the remaining step of generating the nowcast is extrapolation.

When the cascade decomposition and the autoregressive model are used for scale filtering (the S-PROG model), the additional steps include those marked with green color in Fig. 4. When generating ensembles (the STEPS model), stochastic perturbations are added to the AR(p) models and to the advection field using the methods implemented in the `noise` module.

These steps are marked with blue color in Fig. 4.

The ensemble generation is parallelized by using the dask library. For each time step, this is done by splitting the computation to the available processor cores so that each core is responsible for computation of one ensemble member.

Given the input radar composites and the motion field, all operations involved in generating a nowcast are called from the `nowcast` module, except optional post-processing. This can be done either by supplying the requested method to the now-

cast generator or separately by using the functionality implemented in the `postprocessing` module. The post-processing includes methods to ensure that the nowcasts have the same statistical properties of the observations (see Sec. 2.8), as well as methods for generating different products, such as ensemble mean or exceedance probabilities of given intensity thresholds. Computation of accumulations from instantaneous rain rates can be done by using the methods implemented in the `utils.dimension` module. Finally, the nowcasts or nowcast ensembles can be verified and plotted by using the `verification`

and `visualization` modules, respectively.





## 4   Evaluation of nowcast quality

Verification is an essential step of forecasting, not only to monitor forecast performance over time, but also to provide feedback on how to improve the model itself (diagnostic verification). For an ensemble forecast, it is necessary to check whether it is unbiased, has the correct dispersion, and that the forecast probabilities are reliable and sharp (e.g. Jolliffe and Stephenson,
2003). In this section, we evaluate these attributes of pysteps ensemble nowcasts using radar composites from Switzerland and Finland, while data from the United States and Australia will also be used in Sec. 5 and Sec. 6.

### 4.1   Description of the data

As of 2019, the radar network operated by the FMI consists of 10 polarimetric C-band Doppler radars. After clutter filtering, the measured radar reflectivities are interpolated into a grid with spatial and temporal resolutions of 1 km and 5 minutes,
respectively. The correction for the vertical profile of reflectivity (VPR) is applied in order to reduce range-dependent biases (Koistinen and Pohjola, 2014). Finally, reflectivities are converted to rainfall intensities using the Z-R relation $Z = 223R^{1.53}$ adapted to the Finnish climate conditions (Leinonen et al., 2012). Ten precipitation events from Finland containing both stratiform and convective precipitation were chosen for this study (Table 7).

The latest 4th generation MeteoSwiss network consists of 5 polarimetric C-band Doppler radars (Germann et al., 2015). The
quantitative precipitation estimation (QPE) product used in this study includes automatic hardware calibration, clutter filtering, correction for beam shielding, correction for VPR effects, Z-R relation $Z = 316R^{1.5}$, and bias adjustment (Germann et al., 2006a). The radar composite is calculated on a 1 km grid every 5 minutes. Ten events consisting of predominantly convective precipitation were chosen from the Swiss data (Table 8).

The US data set comprises the radar mosaics provided by Warning Decision Support System–Integrated Information (WDSS-
II Lakshmanan et al., 2006, 2007), covering the continental United States at a spatial resolution of approximately 1 km. For the WDSS data, the resolution of the precipitation fields is upscaled from 1 km to 4 km by averaging 4x4 grid points to reduce the computational requirements. The chosen precipitation events are described in Table 9.

The radar network operated by the Australian Bureau of Meteorology (BoM) consists of 66 radars, mostly C-band Doppler radars, with S-band polarimetric Doppler radars operating at four major cities. Raw reflectivity observations are quality con-
trolled in real-time to remove non-meteorological echoes and estimate the reflectivity at the earth surface. This equivalent reflectivity at surface is converted into an instantaneous rainfall rate by use of power-law functions tuned on a per radar basis. Finally, rainfall depths are estimated by adjusting the bias of instantaneous rainfall rates based on observations at real-time gauge locations. The QPE grids are calculated with a spatial resolution of 0.5 km every 5 minutes. The BoM radar dataset comprises two precipitation events: a tropical cyclone in northern Australia and a severe convective event in Sydney (Table 10).
Table 4 summarizes the different data sources and resolutions.



## 4.2 Verification metrics

Pysteps includes a number of verification metrics to help users to analyze the general characteristics of the nowcasts in terms of consistency and quality (or goodness). Probabilistic forecasts have been verified using the ROC curve, reliability diagrams, and rank histograms, as implemented in the `verification` module of pysteps.

The Relative Operating Characteristic (ROC) curve (Jolliffe and Stephenson, 2003) measures the ability of a probabilistic forecast to discriminate between precipitation and no precipitation exceeding a given intensity threshold. For a set of probability thresholds, the ROC curve is constructed by plotting the probability of detection (POD) against the false alarm rate (POFD), not to be confused with the false alarm ratio (FAR). For a perfect forecast, the curve passes through the upper left corner (i.e. 100% hit rate and 0% false alarm rate). The area under the ROC curve can be used as a measure of potential skill.

The reliability diagram (Bröcker and Smith, 2007) measures the bias (reliability) and resolution of a probabilistic forecast. For a given intensity threshold, the diagram shows the forecast probability against the observed frequencies, where the probability range $[0,1]$ is divided into $n$ bins. For a reliable forecast, the curve lies on the diagonal. The reliability diagram is often accompanied with a histogram showing the sample size in each bin (sharpness diagram). A sharp forecast has few samples in the middle of the histogram and many on the sides (probability of either 1 or 0).

The rank histogram (Hamill, 2001) measures how well the ensemble spread corresponds to the observed uncertainty. For each nowcast grid pixel, the ensemble members are ranked in increasing order. A pooled histogram is computed by assigning each verifying observation a bin which it falls into among the ensemble members. The first and last bin are assigned for observations below or above all members, respectively. For a forecast ensemble whose distribution is consistent with the observations, the histogram is flat and no observations fall into the first or last bin. To handle ties (e.g. when both the observed precipitation and

several ensemble members are equal to 0), we implemented the method of Hamill and Colucci (1997). The method randomly chooses a bin between $(M + 1)$ and $(M + M_{tied}) + 1$, where $M$ is the number of members smaller than the observation and $M_{tied}$ is the number of ties (ensemble members equal to the observation).

     An additional metric that can be derived from rank histograms is the outlier percentage (OP). The OP measures the proportion of observations falling outside the ensemble, defined by

$$OP = \frac{h_1 + h_{n+1}}{\sum_{i=1}^{n+1} h_i},$$
(16)

where $h_i$ denotes the $i$-th bin of the rank histogram.

     Pysteps also includes standard neighbourhood verification methods, such as the fractions skill score (FSS). FSS provides an intuitive assessment of the dependency of skill on spatial scale from high-resolution precipitation forecasts (Mittermaier and Roberts, 2010). The FSS is computed by comparing the forecast and observed fractional coverage of precipitation exceeding

certain thresholds in spatial windows (neighborhoods) of increasing size. Using FSS it is possible to determine how the forecast skill varies with neighborhood size, and then determine the smallest scale that provides a sufficiently skillful forecast.





## 4.3 Verification results

The quality of ensemble nowcasts produced by pysteps was verified by using the MeteoSwiss data and the default configuration listed in Table 5. Using the reliability diagram, ROC curve and rank histogram as verification metrics, the results of the experiments are shown in Figs. 5-7. The results obtained by using the FMI data were very similar, and thus not shown here.

Figure 5 shows that for the 0.1 mm h$^{-1}$ intensity threshold, reliable and sharp nowcasts can be obtained up to two hours. The ROC area remains over 0.85, and the deviation of the reliability diagrams from the diagonal remains below 0.25. However, there is a noticeable loss of sharpness after three hours. In addition, the curved shape of the reliability diagrams indicates that the pysteps nowcasts are slightly overconfident (Tippett et al., 2014).

When a higher 5 mm h$^{-1}$ intensity threshold is used, Fig. 6a shows a significant deviation of the reliability diagram from the diagonal only after 45 minutes, which is accompanied with loss of sharpness. However, the ROC area remains above 0.8, indicating potentially useful skill. This suggests that more reliable nowcasts could be obtained by implementing additional calibration procedures in a future version of pysteps. Another observation that suggests lack of calibration is that the optimal nowcasts for precipitation/no precipitation are obtained by choosing a very low probability threshold (for a well-calibrated nowcast this would be 0.5).

The rank histograms (Fig. 7) also show some ensemble under-dispersion with larger values on the first and last bins. In general, we found that there are more misses than false alarms (i.e. cases when all members are lower than the observations). This occurs, for instance, in cases of convective initiation. Despite the ability of pysteps to generate some new light random rain, it is not designed to represent the uncertainty related to an explosive initiation of a thunderstorm.

## 4.4 Numerical diffusion analysis

Conventional semi-Lagrangian schemes are implemented in a recursive way so that the precipitation intensities are interpolated at each time step, which usually leads to substantial numerical diffusion (i.e. loss of power at high spatial frequencies). In the pysteps method (the `extrapolation.semilagrangian` module), this is done by iteratively tracing the locations of precipitation parcels and interpolating the intensities only as the final step of the advection (Germann and Zawadzki, 2002).

To verify the advantage of this implementation, we computed radially averaged Fourier spectra of deterministic nowcasts at various lead times for FMI event no. 3 (Fig. 8). The analysis is performed using the three optical flow methods to understand whether the semi-Lagrangian scheme is sensitive to quality of the motion field. Figure 8 shows an almost perfect overlap of the forecast and observed spectra, an indication that the numerical diffusion of the semi-Lagrangian scheme is very low. Several cases have been analyzed and provided similar results (not shown).

## 4.5 Spatial structure analysis

The aim of probabilistic nowcasting is to generate a reliable ensemble of equiprobable realizations of precipitation fields characterized by power spectra similar to those in Fig. 8. Fig. 9a shows the average spectra of a stochastic 48-member nowcast for





the FMI precipitation event no. 3. Despite a small loss of power for scales ($<$ 100 km), all spectra are close to the observations. In other words, the spatial structure of ensemble members remains realistic at all forecast lead times.

Figure 9b shows the results of the same analysis, but for the ensemble mean forecast (the average of ensemble members). The process of ensemble averaging should produce precipitation fields that become smoother with lead time, which is the

aftermath of the loss of predictability at small scales (Surcel et al., 2014). As expected, Fig. 9b shows a gradual loss of power at small scales. The departure of the forecast spectra from the observed ones occurs at increasing wavelengths, i.e. $\approx$16 km at 5 min and $\approx$128 km at 60 min. However, after 30 min there is a certain increase of power at wavelengths smaller than 16-32 km. This behavior is attributed to the limited ensemble size, which is not large enough to filter out precipitation features at small scales. Thus, one may argue that if the ensemble is too small to model the loss of predictability at such scales, it may also be

too small to reliably model the forecast uncertainty.

An alternative way to deterministically represent the forecast uncertainty is to filter out the unpredictable features using the S-PROG model (Fig. 9c). Also in this case, the departures of forecast spectra from the observed one occur gradually as in the ensemble mean. The first two lead times are remarkably similar, while for lead times beyond 30 min the S-PROG filtering is stronger (at small spatial wavelengths). Again, this level of filtering could be reached with an ensemble of infinite size.

The previous result suggests that we could exploit the discrepancies between the S-PROG and ensemble mean spectra to obtain an estimate of the required ensemble size (as a function of spatial scale and lead time). If the two spectra are similar, it is an indication that the ensemble is large enough.

### 4.6  Temporal structure analysis

To demonstrate the effectiveness of the hierarchy of AR(2) models in modeling the temporal evolution of precipitation, we

derived the theoretical ACF from the the estimated AR parameters (see Eq. 7). The obtained ACF is compared to the empirical ACF between the nowcasts and the corresponding observations. The correlation coefficients are computed separately for each cascade level obtained using the bandpass filters shown in Fig. 1.

Figure 10 shows the average theoretical and empirical ACFs for all the FMI cases. It clearly indicates that the AR(2) process gives accurate estimates of the temporal auto-correlations up to three hours. For smaller scales (0-35 km) having short

lifetimes, the estimates coincide nearly exactly with the observed ones, but for larger scales the auto-correlations are slightly overestimated. This is due to the relatively short memory of the AR(2) process compared to the precipitation lifetimes at these scales (over two hours).

### 5  Sensitivity analysis

The objective of this section is to analyze the sensitivity of pysteps to its configuration options and parameters such as the optical

flow method, the ensemble size, the parameter localization and the cascade decomposition. The default pysteps configuration used in Sec. 4 is based on the results presented here.



## 5.1 Optical flow and scale filtering

Determination of the advection field by optical flow is a key component of any extrapolation-based nowcasting system. Pysteps allows to easily analyze the impact of the optical flow method and also scale filtering on the forecast skill. Moreover, the three methods currently available in the `motion` module constitute an ideal testbed as they cover three very distinct approaches, see the references in Sec. 2.1 for details. The experiments were done by using the MeteoSwiss and US precipitation events described in Tables 8 and 9.

Each optical flow method was used with two deterministic nowcasting methods: a simple extrapolation-based method and S-PROG, which incorporates a scale filtering procedure as described in Seed (2003). Both methods are available in the `nowcasts.extrapolation` and `nowcasts.sprog` modules, respectively. The VET and Lucas-Kanade methods use 2 input images, while DARTS uses 9 input images. The forecast quality was evaluated using the critical success index (CSI) and the mean absolute error (MAE) as described in Jolliffe and Stephenson (2003).

The results of the experiments are shown in Fig. 11. First of all, large differences between the simple extrapolation and S-PROG nowcasts are observed, which is mainly due to the scale filtering implemented in S-PROG (see Sec. 4.5). For the MeteoSwiss events, applying the filtering improves both CSI and MAE, especially at longer lead times (Figs. 11a and 11c). After two hours, the S-PROG nowcasts show a ∼20% increase in the CSI and and ∼40% reduction in the MAE. A similar behavior is observed for the US events (Figs. 11b and 11b) but with a ∼20% the reduction in MAE after two hours.

On the other hand, no significant differences can be observed between different optical flow methods (less than 2%), with DARTS performing slightly worse than the other methods. This is possibly explained by the fact that, with the default configuration, DARTS produces only a large-scale approximation of the advection field.

Figure 12 shows advection fields obtained using different optical flow methods for a selected case (US, 2013/04/11 0800 UTC). Lucas-Kanade and VET produce smooth fields that are remarkably similar, particularly close to the precipitation areas (Figs. 12a and 12b). Within precipitation areas, DARTS produces similar motion fields than the other two methods, but outside precipitation the fields are considerably different.

We also measured the computation times of different optical flow methods in the MeteoSwiss and FMI domains, and the results are shown in Table 6. The experiments were done using an Intel Xeon E5645 CPU with 12 cores running at 2.4 GHz with parallelization enabled in the optical flow methods. The results reflect the fact that the Fourier space and local methods (DARTS and Lucas-Kanade) have significantly lower computational requirements than variational methods (VET), which are however still within the needs of a real-time operational system. Thus, our conclusion from the results shown in Fig. 11 and Table 6 is that the choice of the optical flow method plays a less significant role while nowcast errors are more clearly determined by the dynamic scaling properties of precipitation as highlighted by the large impact of scale filtering on the forecast skill.

## 5.2 Ensemble size

The ensemble size is one of the main factors contributing to the quality and computation time of pysteps nowcasts, and one has to make trade-off between these two. To determine the optimal number, the skill of the nowcasts with different intensity



thresholds and ensemble sizes was evaluated by using two metrics. These are the area under the ROC curve and the outlier percentage (OP). The results are shown in Figs. 13 and 14.

Figure 13 shows that the choice of the ensemble size plays a significant role, which is particularly true when nowcasts of higher precipitation intensities are desired. Figure 13a shows that for $n = 6$, the ROC area falls below 0.85 after two hours,

while it is close to 0.9 when $n$ is increased to 48. However, there is only marginal improvement when $n$ is increased from 24 to 48, which suggests that 24 members is sufficient when nowcasts of precipitation/no precipitation are desired with low intensity thresholds (e.g. 0.1 mm h$^{-1}$). On the other hand, Fig. 13b shows that when the threshold is increased to 5 mm h$^{-1}$, a significant improvement can be expected when increasing $n$ from 48 to 96 or even over 100.

The OP is highly dependent on the ensemble size, which can be observed from Fig. 14. With 96 ensemble members, OP is

below 15% after 20 minutes, which indicates that the ensembles are well able to capture the uncertainties in the spatiotemporal evolution of precipitation. The OP could be further reduced by increasing the ensemble size over 100. Another observation from Fig. 14 is the significant dependence of OP on the lead time. Highest OP can be observed at 20 minutes, and after three hours it is up to 50% smaller.

We also analyzed the computation times needed to generate nowcast ensembles. In a real-time setting it is essential to

know how many ensemble members can be produced before the arrival of the next input radar rainfall image (usually every 5 minutes). To this end, one-hour nowcasts were computed with different ensemble sizes and number of parallel threads using the FMI and MeteoSwiss data listed in Tables 7 and 8, respectively.

The results of the above experiments are shown in Fig. 15. Fig. 15a shows that for the input grid of 710x640 pixels used in the MeteoSwiss domain, it is possible to generate one-hour nowcast ensembles of up to 48 members in less than two minutes

using a server with 12 processor cores.

The results for the larger FMI domain with grid size of 760x1226 pixels are shown in Fig. 15b. Compared to the MeteoSwiss domain, the height of the grid is doubled, which also doubles the computation time (the computational complexity increases quadratically with respect to grid size). Nevertheless, using 12 processor cores, the computation time of a 48-member ensemble still remains below two minutes.

In addition, Figs. 15a and 15b show the effectiveness of the parallelization scheme implemented in pysteps. That is, when plotted in logarithmic scale, the computation time decreases approximately linearly with respect to the number of threads (i.e. the computation time is halved when the number of threads is multiplied by two).

### 5.3 Localization

This experiment investigates the impact of localization on the nowcast quality. For this purpose, we seek at using a spatial subset

of the observations in order to estimate the model parameters so that they can vary in space. The short-space approach presented in Nerini et al. (2017) is generalized to the whole nowcasting system (see module `nowcasts.sseps`). Conceptually similar to the approach in Sideris et al. (2018), the method essentially implements a moving-window localization of the nowcasting procedure, whereby all parameters are estimated over a subdomain of prescribed size. The localization is applied to the cascade decomposition, the autoregressive process (5), the nonparametric Fourier filter (10) and the probability matching (15).





The impact of localization is assessed in terms of rank histograms and reliability diagrams (threshold of 1.0 mm h$^{-1}$) for a 30-minute lead time (Fig. 16). The localization shows positive effects in the ensemble spread, which improves both in terms of reliability and conditional bias, although we also observe a slight decrease of sharpness. This is reflected in the rank histograms, which tend to get flatter as the localization window gets smaller. This seems to be mainly driven by a reduction in the proportion

of observations lying above the ensemble, which reduces from approximately 13% to 8%.

### 5.4  Cascade decomposition

The cascade decomposition was designed to account for dynamic scaling (i.e. the dependence of predictability on spatial scale, see Sec. 2.4). Without the decomposition, precipitation fields are expected to evolve with the same rate at all spatial scales following a single AR process. In such case, the persistence of small- (large-) scale precipitation features would be overesti-

mated (underestimated). Thus, our main hypothesis is that dynamic scaling properties are necessary to produce ensembles with realistic temporal evolution and dispersion of precipitation across spatial scales.

To test our hypothesis, we compared the stochastic nowcasts (`nowcasts.steps` module) with and without cascade decomposition, that is, using 8 or 1 cascade levels, respectively. The objective is to analyze the realism of the temporal evolution, not whether the AR is an appropriate model of the forecast error as in Sec. 4.6. In practice, this implies comparing the theoreti-

cal ACFs of forecast and observed fields as follows: 1) generate nowcasts with either an 8-level or 1-level cascade, 2) transform the forecast fields into the Lagrangian frame (by using the same motion field estimated at start time), 3) decompose the forecast fields into a 6-level cascade, 4) estimate the AR(2) parameters at each scale, 5) derive the full temporal auto-correlation function (ACF, see also Fig. 10), and 6) integrate the ACF to estimate the precipitation lifetime. The procedure is repeated for each forecast lead time up to 2 hours and also for the corresponding observations.

Figure 17 shows the average lifetime for all the MeteoSwiss events plotted against spatial wavelength (in loglog scale). As expected, the model with 8 cascade levels reproduces well the dynamic scaling properties, especially at small wavelengths. However, there is some degree of overestimation of the lifetime at large wavelengths compared to the observations. One possibility would be to adjust the AR parameters to obtain faster decorrelation, thus shorter lifetime, at such scales.

The model without cascade decomposition compensates for the overestimation of persistence at large wavelengths, but

strongly overestimates the one of small wavelengths. Hence, the evolution of convective cells in the stochastic nowcast is too slow compared with reality. This could be checked visually by looking at nowcast animations with and without decomposition (https://github.com/pySTEPS/pysteps-publication/tree/master/animations).

Another approach to understand the impact of the cascade decomposition is to analyze the filtering properties of the ensemble mean forecast (e.g. Surcel et al., 2014). Figure 18 illustrates the evolution of the ensemble mean forecast spectra with 8 and 1

cascade levels, respectively. When using the cascade decomposition the process of ensemble averaging leads to a loss of power at small spatial wavelengths, in agreement with the expected loss of predictability (see Sec. 4.5). Instead, the model with one cascade level is not able to filter out the unpredictable features. As a consequence, it may not be able to adequately characterize the loss of predictability (and uncertainty) at different spatial scales.





Figure 19 illustrates the ensemble and probabilistic verification for all the MeteoSwiss events with and without cascade decomposition. The sensitivity of forecast uncertainty estimations on using the incremental precipitation mask is also included.

The rank histograms behave differently depending on the chosen forecast settings (Fig. 19a). The two models without decomposition denote a clear overdispersion with a characteristic dome-shape in the bin range 13-22, especially for the setting 5 with 1 level and no mask. Instead, the models with 8 levels display a flat histogram, except for the very last bin, which contains the frequency of observations exceeding all the ensemble members (misses). The last bin is also quite sensitive to using the mask, which prevents the ensemble to capture the uncertainty associated to precipitation initiation far from the main precipitation body.

Figure 19b shows the spread-error relationship analysis (i.e. the standard deviation among all ensemble members) against 10 the average RMSE of all members against the observations. The experiments with 8 levels have both a lower RMSE and spread than the ones using 1 level. It can also be noticed that the 1-level models do not show the same overdispersion that was observed on the rank histograms.

Finally, the reliability diagrams of Fig. 19c-d demonstrate a very good reliability for all forecast settings, although the forecast probabilities of the models with 1 level are slightly lower than the observed frequencies. In addition, the 8-level model 15 has better sharpness, i.e. a larger proportion of high forecast probabilities ($> 0.9$).

# 6   Nowcasting the extremes: two severe-weather case studies from Australia

An example of applying the pysteps library in order to forecast rainfall fields for Tropical Cyclone Penny and Severe Convection in Sydney (Australia) is shown in Figs. 20 and 21 respectively. The ability of pysteps to estimate diverse advection patterns from observed data is quite clear in these examples, with the Tropical Cyclone case showing a clear clockwise rotational pattern 20 while the Severe Convection shows an almost even easterly flow pattern across the whole domain. Tropical Cyclone nowcasts preserve the original cyclonic pattern up to 60 minutes ahead but some distortions are induced for longer lead times due to convergence and divergence. The Severe Convection case has a simpler advection pattern that helps to preserve the general structure of the observed rainfall fields beyond 60 minutes. Additional data sources such as satellite or NWP forecasts may help to estimate future advection velocities and reduce potential anomalies for longer lead times. It is important to note however 25 that post-processing of nowcasts (see Section 2.8) ensures that the forecast rainfall fields have the same statistical properties with the observed ones in both case studies.

## 6.1   Neighbourhood verification

Figures 22 and 23 show examples of FSS results calculated by pysteps for different forecast times for both Australian case studies.

30 The FSS decays in both case studies when spatial scale is reduced or when the intensity threshold is increased, although differences exist between the two case studies. For example, the Tropical Cyclone case seems to have a less acute reduction in the skill with changes in spatial scale. This can be related to the presence of a more uniform rainfall distribution across the



domain (large bands of rainfall moving in an organized way) limiting displacement errors at small scales. Instead, the skill reduces heavily as rainfall intensity increases. This drop in skill could have been accentuated by the relatively low number of high intensity samples in these events.

On the other hand, the Severe Convection case displays a stronger decay of skill when spatial scale is reduced, probably due
to the presence of sharp spatial gradients and isolated convective cells. This said, it is interesting to note how for the higher intensities and large spatial scales the FSS values do not decay as heavily as seen in the other case study. This difference could be a consequence of having more high intensity values in the Severe Convection event.

## 6.2 Lifetime of rainfall fields per spatial scale

To compare the behaviour of the AR(2) model for these two case studies, temporal auto-correlation functions for each spatial
scale were calculated using Eq. 7, and then integrated to estimate the precipitation lifetimes for each scale and run time. Figure 24 summarizes the precipitation lifetime results for each case study. Overall, a more diverse set of spatial and temporal patterns observed during the Severe Convection event makes interquartile ranges of precipitation lifetimes larger for this case study for all scales. In comparison, similar organized patterns were present during most of the duration of the Tropical Cyclone event and therefore precipitation lifetime values have a narrower range. Smaller scales seem to have similar average lifetime values
for both cases with no strong temporal variations within the events. For the larger scales, however, precipitation lifetime values for Tropical Cyclone event are greater than Severe Convection ones, again as a consequence of large-scale organized patterns observed in this event.

From an operational perspective, these results illustrate the importance of using an AR(2) model with parameters continuously adjusted to the latest observed patterns to adequately simulate rainfall nowcasts instead of using fixed, historical
parameters. However, it is important to note as well that a number of outliers were obtained in both cases (mainly for the larger spatial scales). These anomalous values may indicate the need of introducing a temporal smoothing scheme during the estimation of the AR(2) parameters. Having a more stable, slowly evolving parameters would help to (i) reduce the possibility of generating unrealistic nowcasts from one particular set of observations and also (ii) create smooth transitions between consecutive rainfall nowcasts.

## 25  7   Conclusions

Pysteps is an open-source library for radar-based probabilistic precipitation nowcasting written in Python. It represents a community-based initiative that aims at connecting nowcasting scientists by sharing code, methods, ideas and results and also to provide an easy-to-use tool for operational applications.

Pysteps implements the main components of an ensemble precipitation nowcasting system. These are input/output, optical
flow and extrapolation routines, time series methods for modeling the temporal evolution of precipitation fields, stochastic noise generation in space and time, visualization and forecast verification.





The development of pysteps is done by using a distributed version control system, and the project is hosted at GitHub (https://pysteps.github.io). The library has a modular design so that developers can easily interchange components and embed them into other software packages.

In this paper, we briefly explained the framework of probabilistic precipitation nowcasting and how such nowcasts can be produced using pysteps. The potential of pysteps was demonstrated using radar composite images from Finland (FMI), Switzerland (MeteoSwiss), United States and Australia (BoM). Finally, we performed experiments, where the the quality of pysteps nowcasts and computational performance were evaluated with different configurations. This brought us to the following conclusions:

1. Probabilistic precipitation nowcasts have good reliability that, however, decreases for increasing rainfall intensity thresholds and lead time. Using the MeteoSwiss data, it was shown that for the $0.1$ mm h$^{-1}$ threshold, reliable nowcasts with potentially useful skill can be obtained up to 3 hours. When the threshold was increased to $5$ mm h$^{-1}$, useful nowcasts could still be obtained up to 45 minutes (Figs. 5 and 6).

2. Rank histograms show that the observed uncertainties in the nowcasts have a good correspondence with the ensemble spread. However, we observed some underdispersion, mostly related to the inability of persistence-based nowcasting to predict the initiation of new convection (misses). However, the proportion of outliers in the rank histogram is only 10-15% with a 24-member ensemble on the MeteoSwiss precipitation events (Figs. 7 and 14).

3. The stochastic ensemble members have realistic spatial and temporal structure, as confirmed by Fourier analysis (Figs. 8, 9, 10 and 17).

4. The three optical flow methods that we tested, i.e. Lucas-Kanade, DARTS and VET, provided similar forecast accuracy (differences less than 2%, see Fig. 11). Thus, the choice of optical flow method is not a first order problem in terms of nowcast quality, although there may be some specific situations requiring more advanced schemes, e.g. in presence of orographic rain and/or multiscale motion. Choosing the fastest optical flow method frees time to generate a larger ensemble. When tested with the FMI and MeteoSwiss data, DARTS and Lucas-Kanade computed the motion field in less than 5 seconds.

5. With parallelization implemented via the dask library, pysteps can generate relatively large ensembles. For example, using 4 CPU cores on the MeteoSwiss grid (710x640), it is possible to produce a 48-member ensemble up to +1h (12 frames) in about 2 min (Fig. 15).

6. Localizing the nowcasting procedure (i.e. the cascade decomposition, noise generation, AR process and probability matching), is beneficial in terms of probabilistic forecast skill (Fig. 16). The need for localization is intuitively important on large domains, where stratiform and convective rain can coexist in different geographical regions, but also on smaller domains characterized by complex orography, as demonstrated in this study. These results highlight the importance of defining an appropriate model domain for pysteps. That is, the one that compromises between the need for homogeneous statistical properties (i.e. a small domain) and the need for a robust estimation of model parameters (i.e. large domain).





7. Considering the scale dependence of precipitation predictability is clearly important. The Fourier-based cascade decomposition provides an adequate framework, which can be easily extended to account for spatial localization (i.e. the short-space FFT). Other decomposition frameworks can be explored, but it is not yet clear whether there is a benefit in terms of forecast quality.

8. In presence of extreme precipitation, pysteps can still deliver skillful nowcasts up to one hour for specific intensity and spatial scales (Figs. 22 and 23). A wide range of predictability is observed between and within the events (Fig. 24), thus highlighting the importance of having an adaptive approach that continuously updates the model parameters in real time.

Our analyses not only helped understanding the importance of certain nowcasting concepts, but were the basis to define a minimum viable product (MVP), which constitutes the default configuration of pysteps (see Tab. 5). Additional levels of complexity (e.g. localization) can be included at the cost of computational time and robustness. Users are responsible for evaluating whether it is worth the effort in terms of forecast quality and computational resources. Also, we advise pysteps users to visualize and verify precipitation nowcasts before using them for other applications, for instance, as inputs to hydrological models.

## 7.1 Potential extensions and applications of pysteps

Pysteps represents a long-term effort that does not end with the publication of this paper. Pysteps already provides a quite comprehensive library, but still misses two important modules: 1) a module to generate QPE ensembles characterizing the radar measurement uncertainty (e.g. Jordan et al., 2003; Germann et al., 2009), and 2) a module for seamless blending of precipitation fields from different data sources, such as radar nowcasts and NWP forecasts (Bowler et al., 2006; Nerini et al., 2019), radar, satellite and NWP data (Renzullo et al., 2017).

It would be interesting to include other state-of-the-art ensemble precipitation nowcasting systems in pysteps, for example PHAST (Metta et al., 2009), SBMcast (Berenguer et al., 2011), SAMPO-TBM (Leblois and Creutin, 2013), and NowPrecip (Sideris et al., 2018). A large-scale forecast verification intercomparison project could be foreseen to better understand the advantages and disadvantages of different ensemble nowcasting techniques.

Pysteps opens a number of possibilities that go beyond the field of nowcasting. The most natural application of pysteps is to use the precipitation ensembles as inputs into hydrological models for uncertainty quantification, in both urban and rural environments (e.g. Zappa et al., 2011; Thorndahl et al., 2017).

Individual pysteps modules could also serve different purposes. For example, the optical flow modules could be used to study precipitation growth and decay in moving coordinates (e.g. Foresti et al., 2018; Zeder et al., 2018), to correct radar field accumulations accounting for advection (e.g. Wang et al., 2015; Lukach et al., 2017), to synchronize the individual radar elevation scans (e.g. Tabary, 2007), or to separate the location error of NWP precipitation forecasts (Marzban and Sandgathe, 2010).

The tools available in the noise and the time series modules could be used for stochastic simulation of design storms (e.g. Seed et al., 1999; Paschalis et al., 2013), weather generators (Peleg et al., 2017), and also to understand and quantify the sub-



pixel variability of radar rainfall (e.g. Gires et al., 2014; Benoit, 2018; Peleg et al., 2016). Other applications could include stochastic downscaling or emulation of climate model output (e.g. Raut et al., 2018; Beusch et al., 2018).

We encourage the nowcasting community and potential users to implement new nowcasting methods, propose new modules, try pysteps on different applications, send us feedback, and contribute to the library for the benefit of everyone.

*Code and data availability.* The pysteps library is available at https://github.com/pySTEPS/pysteps. The scripts to run the experiments and produce the figures of this paper are available at https://github.com/pySTEPS/pysteps-publication. The radar data are available upon request.

**Appendix A: Pysteps use cases**

Listing 1 demonstrates how to browse and read archived radar composites using the `io` module and decompose a radar image into a cascade using the `cascade` module. The desired time stamp, root path of the data archive, the file name pattern and ex-
tension for finding the input file are specified at lines 9-12. Finding the input files is done by using `io.archive.find_by_date` at lines 14 and 15. Nine previous input files preceding the desired time stamp are also retrieved with the given time step of five minutes. The retrieved file names are then supplied to `io.read_timeseries` that returns a three-dimensional array of shape (num_timesteps,height,width) containing the radar precipitation fields and a metadata dictionary (line 19). The cascade decomposition is done by initializing the Fourier filter with `filter_gaussian` and calling `decomposition_fft` (lines
15 24-25).

Listing 2 demonstrates computation of a deterministic S-PROG nowcast and a STEPS nowcast ensemble. This is done in two stages: computation of the advection field using the Lucas-Kanade method implemented in the `motion` module (lines 8-9) and then computing the nowcasts with the motion field supplied as input (lines 12-18). The callable functions for these are retrieved using the `get_method` interface. The inputs for the nowcasting methods are a time series of three radar composites
(Z), the motion field (V), the number of time steps (12), the number of cascade levels (8) and the threshold for rain/no rain (-10 dBR). In addition, for STEPS the ensemble size and the spatial and temporal resolution of the data are set to 24, one kilometer and five minutes, respectively. The resulting nowcasts are shown in Figs. 25 and 26.

Verification of a nowcast ensemble is shown in Listing 3 using three metrics: ROC, reliability diagram and rank histogram (see Sec. 4.2). The verifying observations are imported at lines 4-10 by using the `io` module. This is followed by a loop
over the time steps of the nowcast. For each time step, the verification metric is initialized with the `*_init` function, and the verification data is accumulated to the resulting object by calling `*_accum`, which allows accumulating data from multiple events. In addition, exceedance probabilities (P_f) for the 0.1 mm h$^{-1}$threshold used above are computed by using `ensemblestats.excprob`.

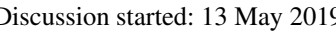



**Listing 1** Read 10 archived radar reflectivity composites in PGM format, apply thresholding and decompose the last one into a 7-level cascade and plot the cascade levels.

```python
from datetime import datetime
from matplotlib import cm, pyplot
import numpy as np
from pysteps.cascade.bandpass_filters import filter_gaussian
from pysteps import io
from pysteps.io.importers import import_fmi_pgm
from pysteps.cascade.decomposition import decomposition_fft
from pysteps.utils import conversion, transformation
date      = datetime.strptime("201609281600", "%Y%m%d%H%M")
root_path = "pysteps-data/radar/fmi"
fn_pattern = "%Y%m%d%H%M_fmi.radar.composite.lowest_FIN_SUOMI1"
fn_ext    = "pgm.gz"
# find the input files from the archive
fns = io.archive.find_by_date(date, root_path, "%Y%m%d", fn_pattern, fn_ext, 5,
num_prev_files=9)
# read the radar composites and apply thresholding
Z, _, metadata = io.read_timeseries(fns, import_fmi_pgm, gzipped=True)
R = conversion.to_rainrate(Z, metadata, 223.0, 1.53)[0]
R = transformation.dB_transform(R, threshold=0.1, zerovalue=-15.0)[0]
R[~np.isfinite(R)] = -15.0
# construct bandpass filter and apply the cascade decomposition
filter = filter_gaussian(R.shape[1:], 7)
decomp = decomposition_fft(R[-1, :, :], filter)
# plot the normalized cascade levels
for i in range(7):
mu = decomp["means"][i]
sigma = decomp["stds"][i]
decomp["cascade_levels"][i] = (decomp["cascade_levels"][i] - mu) / sigma
fig, ax = pyplot.subplots(nrows=2, ncols=4)
ax[0, 0].imshow(R[-1, :, :], cmap=cm.RdBu_r, vmin=-3, vmax=3)
ax[0, 1].imshow(decomp["cascade_levels"][0], cmap=cm.RdBu_r, vmin=-3, vmax=3)
ax[0, 2].imshow(decomp["cascade_levels"][1], cmap=cm.RdBu_r, vmin=-3, vmax=3)
ax[0, 3].imshow(decomp["cascade_levels"][2], cmap=cm.RdBu_r, vmin=-3, vmax=3)
ax[1, 0].imshow(decomp["cascade_levels"][3], cmap=cm.RdBu_r, vmin=-3, vmax=3)
ax[1, 1].imshow(decomp["cascade_levels"][4], cmap=cm.RdBu_r, vmin=-3, vmax=3)
ax[1, 2].imshow(decomp["cascade_levels"][5], cmap=cm.RdBu_r, vmin=-3, vmax=3)
ax[1, 3].imshow(decomp["cascade_levels"][6], cmap=cm.RdBu_r, vmin=-3, vmax=3)
ax[0, 0].set_title("Observed")
ax[0, 1].set_title("Level 1")
ax[0, 2].set_title("Level 2")
ax[0, 3].set_title("Level 3")
ax[1, 0].set_title("Level 4")
ax[1, 1].set_title("Level 5")
ax[1, 2].set_title("Level 6")
ax[1, 3].set_title("Level 7")
for i in range(2):
for j in range(4):
ax[i, j].set_xticks([])
ax[i, j].set_yticks([])
pyplot.savefig("cascade_decomp.png", dpi=300, bbox_inches="tight")
```



---

**Listing 2** Compute the advection field and S-PROG and STEPS nowcasts from the reflectivity composites obtained in Listing 1 and plot the nowcasts. The nowcasts are computed by using 12 time steps (i.e. one-hour nowcast with the 5-minute time step), 8 cascade levels and -10 dBR intensity threshold. The STEPS nowcast is computed by using 24 ensemble members.

---

```python
from matplotlib import pyplot
from pysteps import motion, nowcasts
from pysteps.postprocessing.ensemblestats import excprob
from pysteps.utils import transformation
from pysteps.visualization import plot_precip_field, quiver

# compute the advection field
oflow_method = motion.get_method("lucaskanade")
V = oflow_method(R)

# compute the S-PROG nowcast
nowcast_method = nowcasts.get_method("sprog")
R_f_sprog = nowcast_method(R[-3:, :, :], V, 12, R_thr=-10.0)[-1, :, :]

# compute the STEPS nowcast
nowcast_method = nowcasts.get_method("steps")
R_f = nowcast_method(R[-3:, :, :], V, 12, n_ens_members=24, n_cascade_levels=8,
                     R_thr=-10.0, kmperpixel=1.0, timestep=5)

# plot the S-PROG nowcast, one ensemble member of the STEPS nowcast and the exceedance
# probability of 0.1 mm/h computed from the ensemble
R_f_sprog = transformation.dB_transform(R_f_sprog, threshold=-10.0, inverse=True)[0]
pyplot.figure()
plot_precip_field(R_f_sprog, map="basemap", geodata=metadata, drawlonlatlines=True,
                  basemap_resolution='h')
pyplot.savefig("SPROG_nowcast.png", bbox_inches="tight", dpi=300)

R_f = transformation.dB_transform(R_f, threshold=-10.0, inverse=True)[0]

R_f_mean = np.mean(R_f[:, -1, :, :], axis=0)

pyplot.figure()
plot_precip_field(R_f_mean, map="basemap", geodata=metadata, drawlonlatlines=True,
                  basemap_resolution='h')
pyplot.savefig("STEPS_ensemble_mean.png", bbox_inches="tight", dpi=300)

pyplot.figure()
plot_precip_field(R_f[0, -1, :, :], map="basemap", geodata=metadata, drawlonlatlines=True,
                  basemap_resolution='h')
pyplot.savefig("STEPS_ensemble_member.png", bbox_inches="tight", dpi=300)

pyplot.figure()
P = excprob(R_f[:, -1, :, :], 0.5)
plot_precip_field(P, map="basemap", geodata=metadata, drawlonlatlines=True,
                  basemap_resolution='h', type="prob", units="mm/h", probthr=0.5)
pyplot.savefig("STEPS_excprob_0.5.png", bbox_inches="tight", dpi=300)
```

---

## Appendix B: Precipitation events

The precipitation events to test pysteps come from Finland, Switzerland, USA and Australia. They are described in Tables 7, 8, 9, and 10, respectively.





**Listing 3** Compute and plot ROC curves, reliability diagrams and rank histograms for the STEPS nowcast generated in Listing 2 with different lead times.

```python
from pysteps.postprocessing import ensemblestats
from pysteps.utils import conversion
from pysteps import verification

# find the files containing the verifying observations
fns = io.archive.find_by_date(date, root_path, "%Y%m%d", fn_pattern, fn_ext,
                              5, 0, num_next_files=12)

# read the observations
Z_obs, _, metadata = io.read_timeseries(fns, import_fmi_pgm, gzipped=True,
                                        num_next_files=12)
R_obs = conversion.to_rainrate(Z_obs, metadata, 223.0, 1.53)[0]

# iterate over the nowcast lead times
for lt in range(R_f.shape[1]):
  # compute the exceedance probability of 0.1 mm/h from the ensemble
  P_f = ensemblestats.excprob(R_f[:, lt, :, :], 0.1, ignore_nan=True)

  # compute and plot the ROC curve
  roc = verification.ROC_curve_init(0.1, n_prob_thrs=10)
  verification.ROC_curve_accum(roc, P_f, R_obs[lt+1, :, :])
  fig = figure()
  verification.plot_ROC(roc, ax=fig.gca(), opt_prob_thr=True)
  pyplot.savefig("ROC_%02d.eps" % (lt+1), bbox_inches="tight")
  pyplot.close()

  # compute and plot the reliability diagram
  reldiag = verification.reldiag_init(0.1)
  verification.reldiag_accum(reldiag, P_f, R_obs[lt+1, :, :])
  fig = figure()
  verification.plot_reldiag(reldiag, ax=fig.gca())
  pyplot.savefig("reldiag_%02d.eps" % (lt+1), bbox_inches="tight")
  pyplot.close()

  # compute and plot the rank histogram
  rankhist = verification.rankhist_init(R_f.shape[0], 0.1)
  verification.rankhist_accum(rankhist, R_f[:, lt, :, :], R_obs[lt+1, :, :])
  fig = figure()
  verification.plot_rankhist(rankhist, ax=fig.gca())
  pyplot.savefig("rankhist_%02d.eps" % (lt+1), bbox_inches="tight")
  pyplot.close()
```

*Author contributions.* The core of the pysteps package was mainly written by Seppo Pulkkinen and Daniele Nerini, with many contributions by Andrés Pérez-Hortal. Loris Foresti has promoted and closely followed the design of pysteps from its birth. The previously mentioned authors, and Carlos Velasco-Forero made many analyses and figures for the paper. Urs Germann and Alan Seed supported the project, supervised the science and improved the manuscript.

5    *Competing interests.* The authors do not have any competing interests.





*Acknowledgements.* Seppo Pulkkinen was supported by the Finnish Academy of Science and Letters via the the Foundations' Post Doc Pool. Loris Foresti and Daniele Nerini were supported by the Swiss National Science Foundation Ambizione project *Precipitation attractor from radar and satellite data archives and implications for seamless very short-term forecasting* (PZ00P2_161316). The authors express their gratitude to Weather Decision Technologies for providing access to the United States radar composites. The whole pysteps community is

5    here also acknowledged for the various contributions and discussions to the project.



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



**Figure 1.** Normalized weight functions with corresponding Fourier wavenumbers and spatial scales for the FMI domain. The domain is a 760x1226 grid at 1 km resolution.



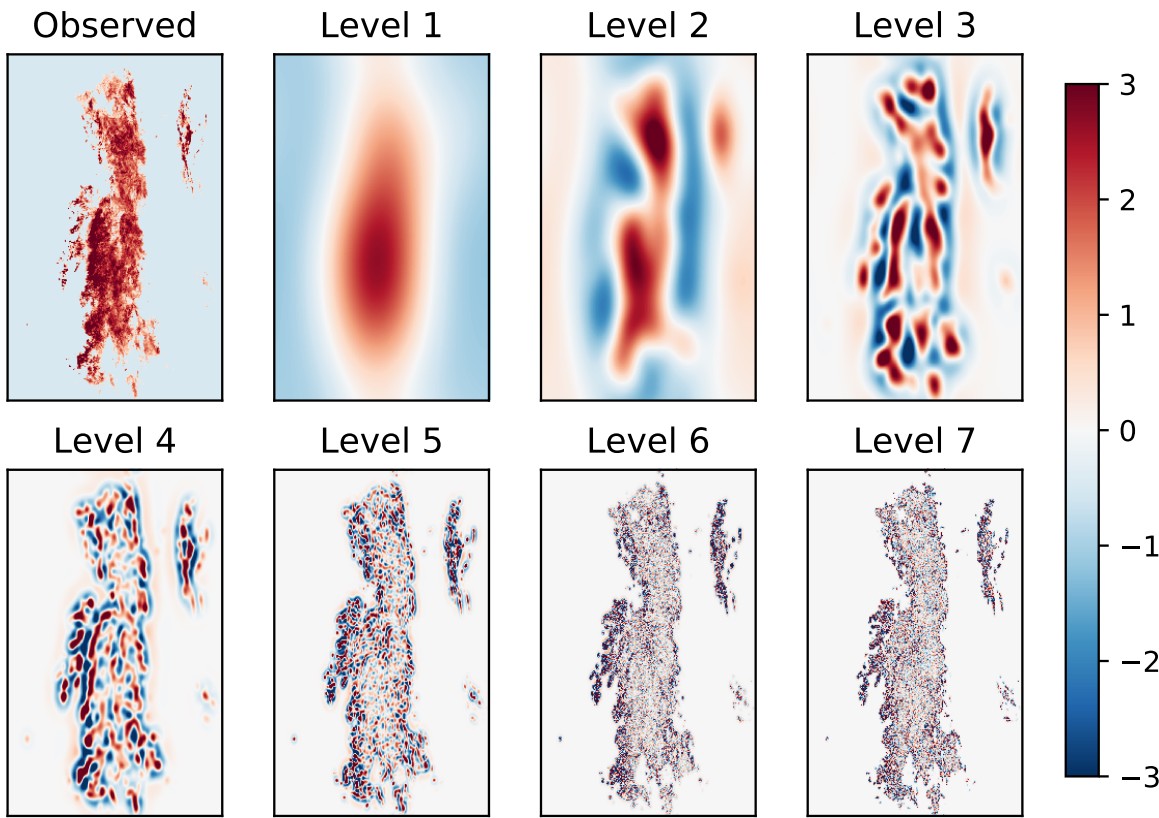

**Figure 2.** The radar observations and 7 first levels of the cascade decomposition of an FMI rain rate composite at time 1600 UTC 28 Sep 2016. Values below -10 dBR were set to -15 dBR before applying the decomposition in order to reduce the discontinuity at the boundaries of precipitation areas. The observed field and the cascade levels have been normalized to zero mean and unit variance. See Listing 1 in Appendix A for obtaining the decomposition.



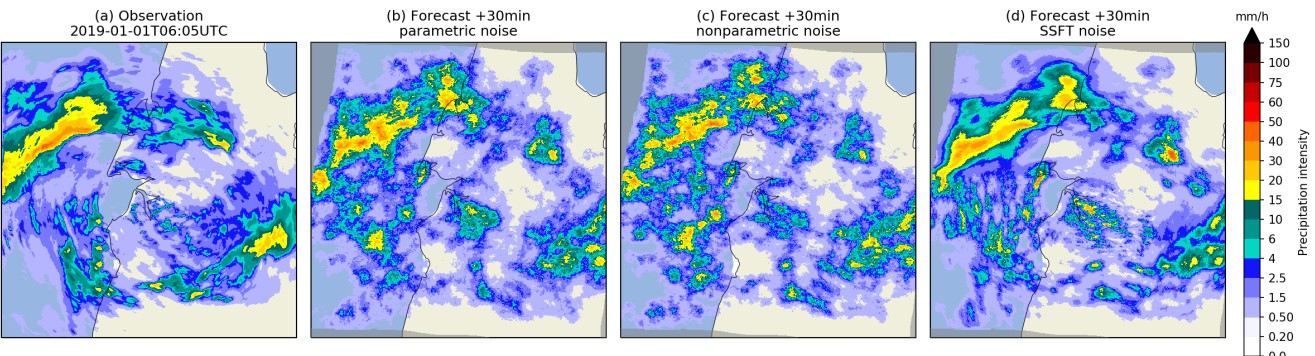

**Figure 3.** Comparison of three +30 minutes stochastic nowcasts produced with the FFT noise generators available in pysteps as described in Sec. 2.6. (a) The radar-based rainfall analysis from the Australian radar network valid at time 0605 UTC 01 January 2019 on a 512x512 pixel grid (event no. 2 in Table 10). (b-d) One member of a +30 minute nowcast produced using (b) the parametric noise generator, (c) the nonparametric generator or (d) the SSFT generator with a 128x128 pixel sliding window. All realizations share the same random seed.


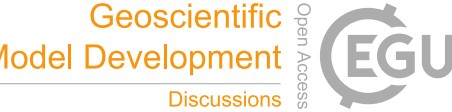

**Figure 4.** Workflow for computing precipitation nowcasts using pysteps. For each chart element, the top row describes the task and the bottom row is the name of the module used for this purpose. White colors represent the operations that are done with all nowcasting methods. Green colors represent the additional operations included when the cascade decomposition and the autoregressive AR(p) model are applied (i.e. the S-PROG model). Finally, blue colors represent the operations that are done when stochastic perturbations are added and the ensemble computation is parallelized (i.e. the full STEPS model).



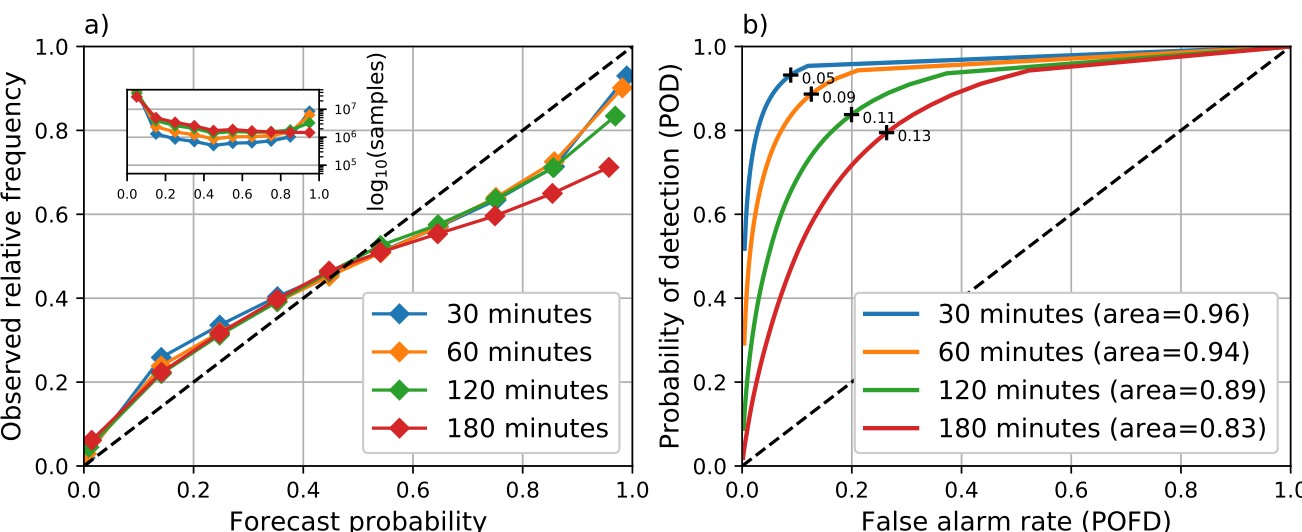

**Figure 5.** Reliability diagrams (a) and ROC curves (b) computed from STEPS nowcasts during the MeteoSwiss events listed in Table 8 with different lead times and threshold 0.1 mm h$^{-1}$. The default settings listed in Table 5 were used for computing the nowcasts. The optimal probability thresholds that maximize POD-POFD are marked in the ROC curves with black crosses.

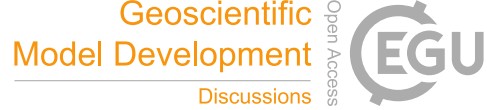



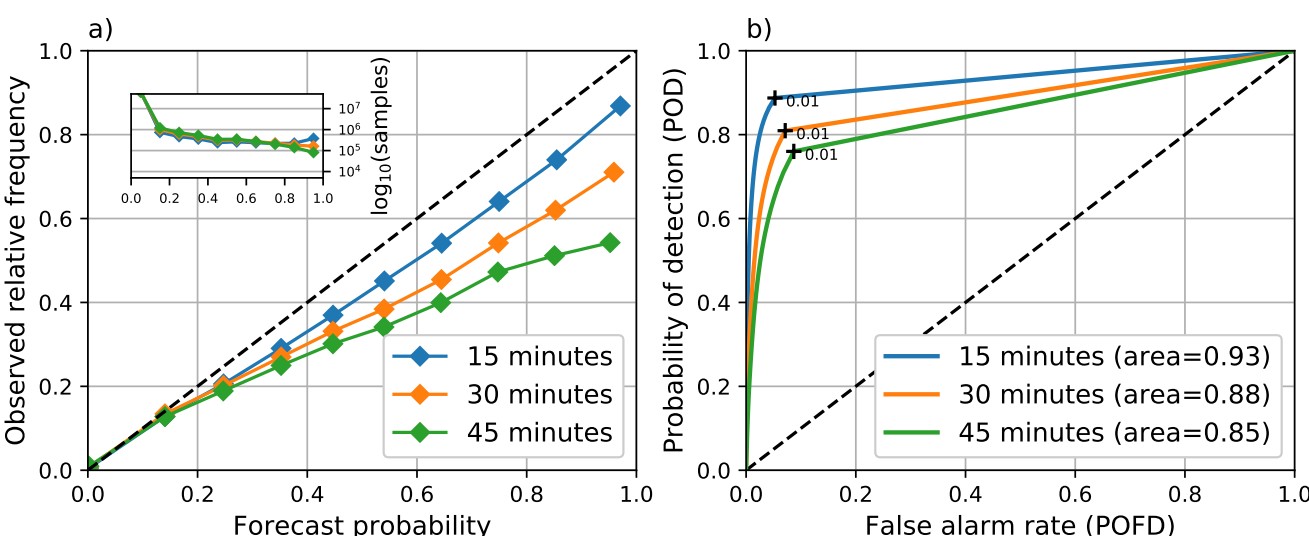

**Figure 6.** Same as Fig. 5, but for an intensity threshold of 5 mm h$^{-1}$.



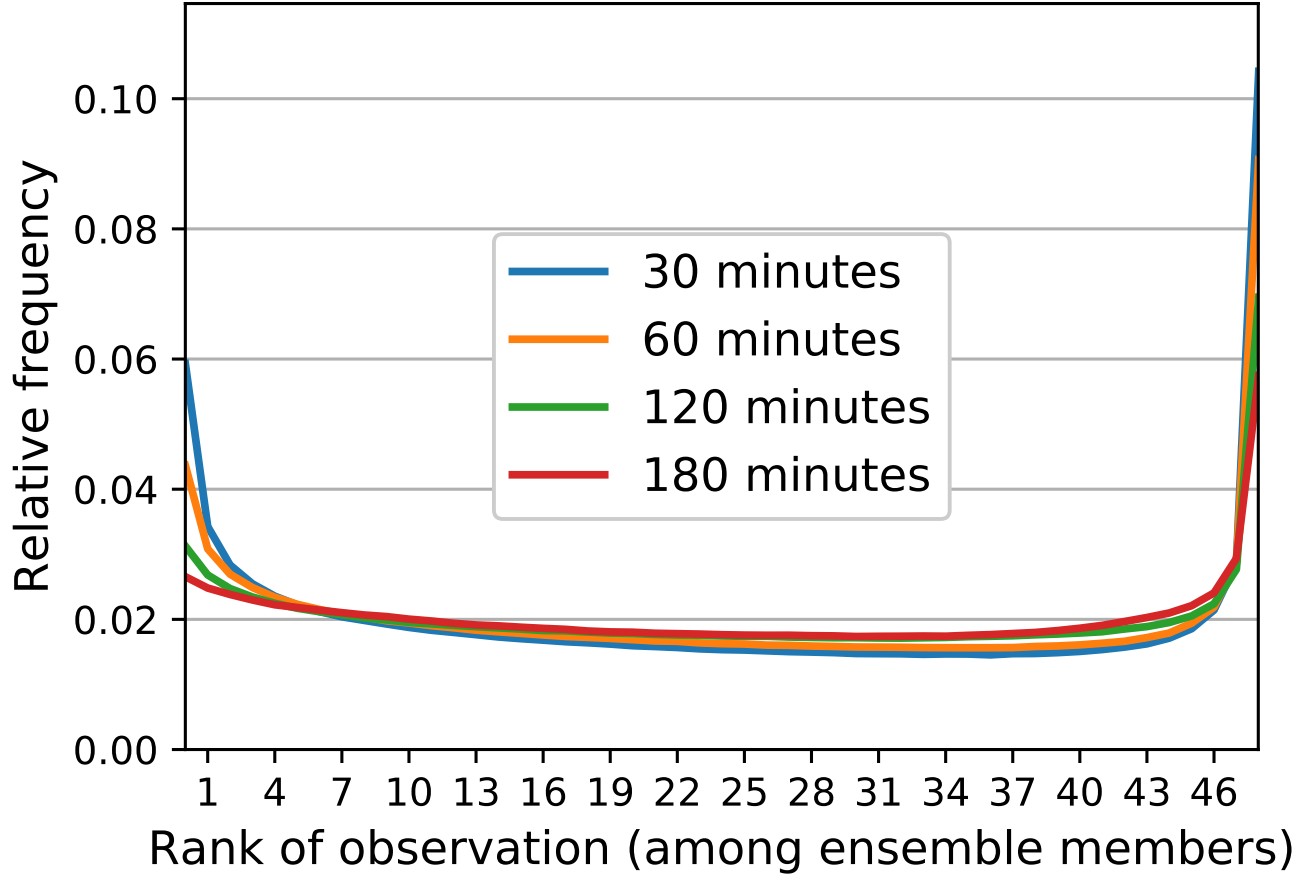

**Figure 7.** Rank histograms computed from STEPS nowcasts during the MeteoSwiss events listed in Table 8 with different lead times. The default settings listed in Table 5 were used for computing the nowcasts.

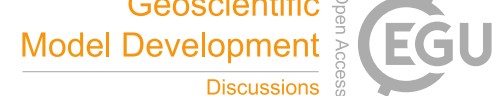



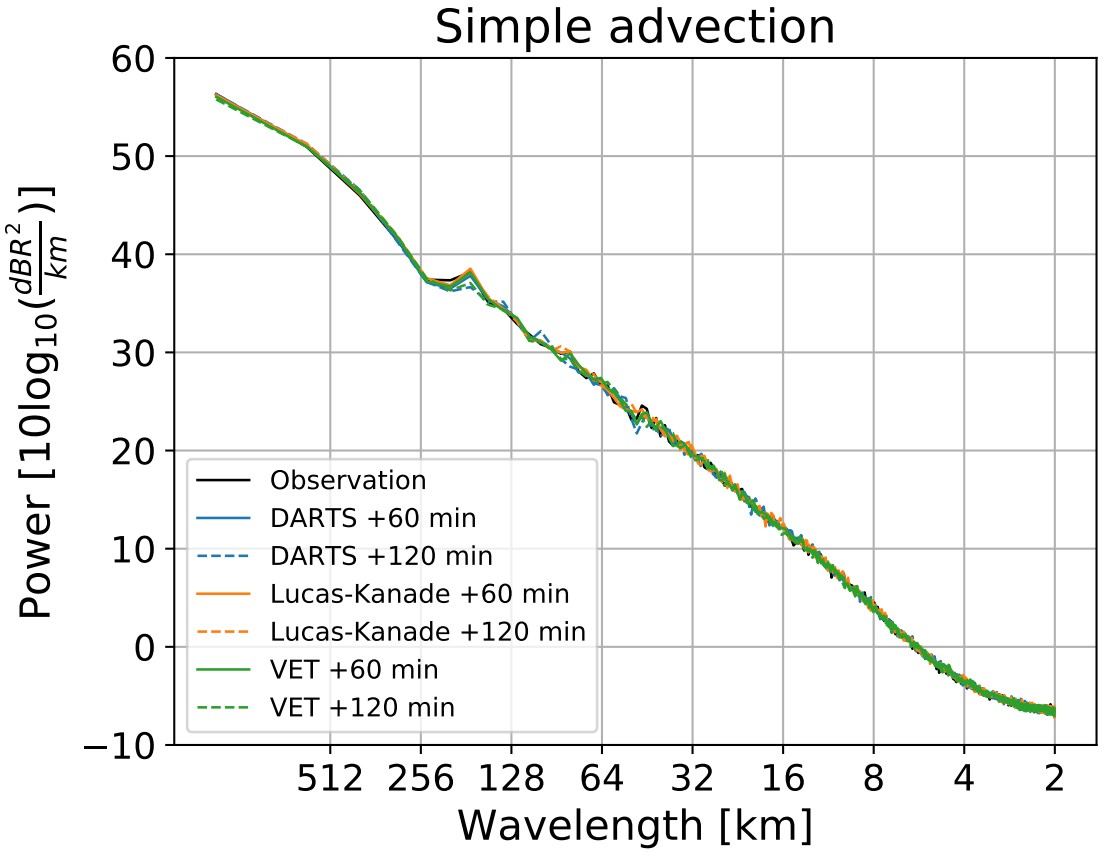

**Figure 8.** Numerical diffusion analysis of the semi-Lagrangian advection scheme using radially averaged Fourier spectra for different optical flow methods and different forecast lead times. The nowcasts are for FMI event no. 3 (1600 UTC 28 Sep 2016).





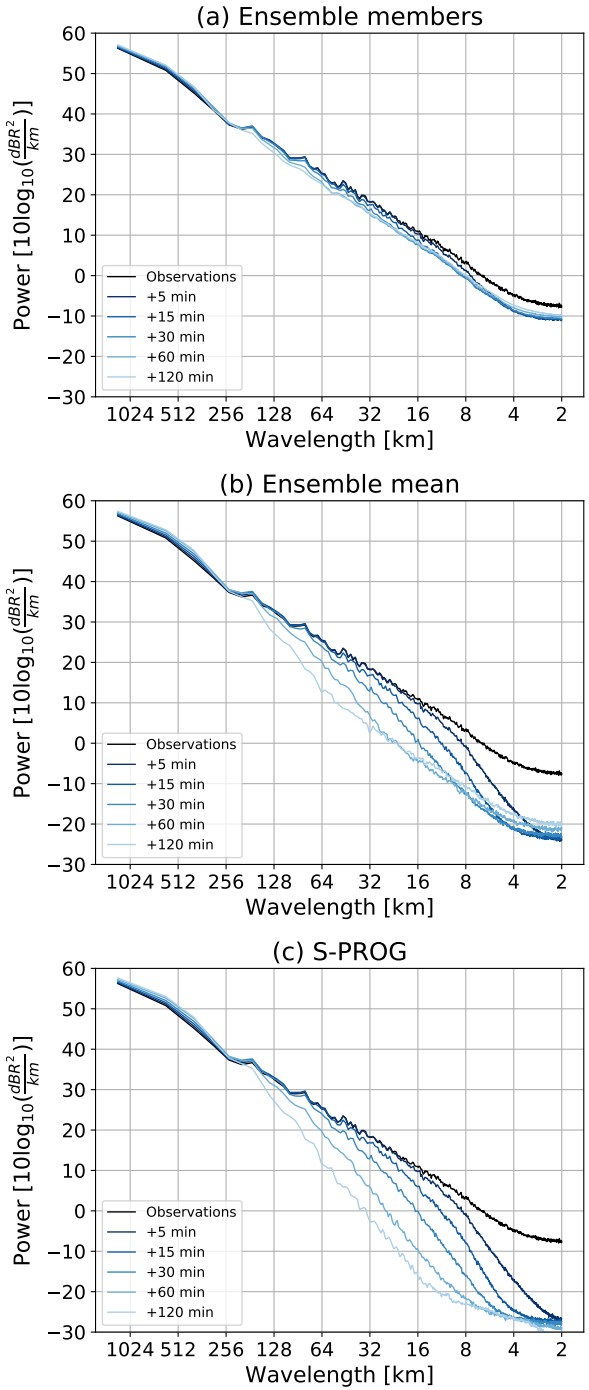

**Figure 9.** Spatial structure analysis of a) stochastic ensemble members, b) ensemble mean, and c) S-PROG filtering. To be comparable, the incremental mask and probability matching were used for both the ensemble mean and S-PROG nowcasts. All nowcasts used the Lucas-Kanade optical flow on the same event of Fig. 8. The ensemble is composed of 48 members. All models used a cascade of 8 levels without motion perturbations.





**Figure 10.** Temporal auto-correlation estimates obtained from AR(2) models (dashed lines) and the correlation between an extrapolation nowcast and the corresponding observations. The analysis is based on the FMI events (Table 7). The line numbers correspond to the frequency bands shown in Fig. 1 from left to right.




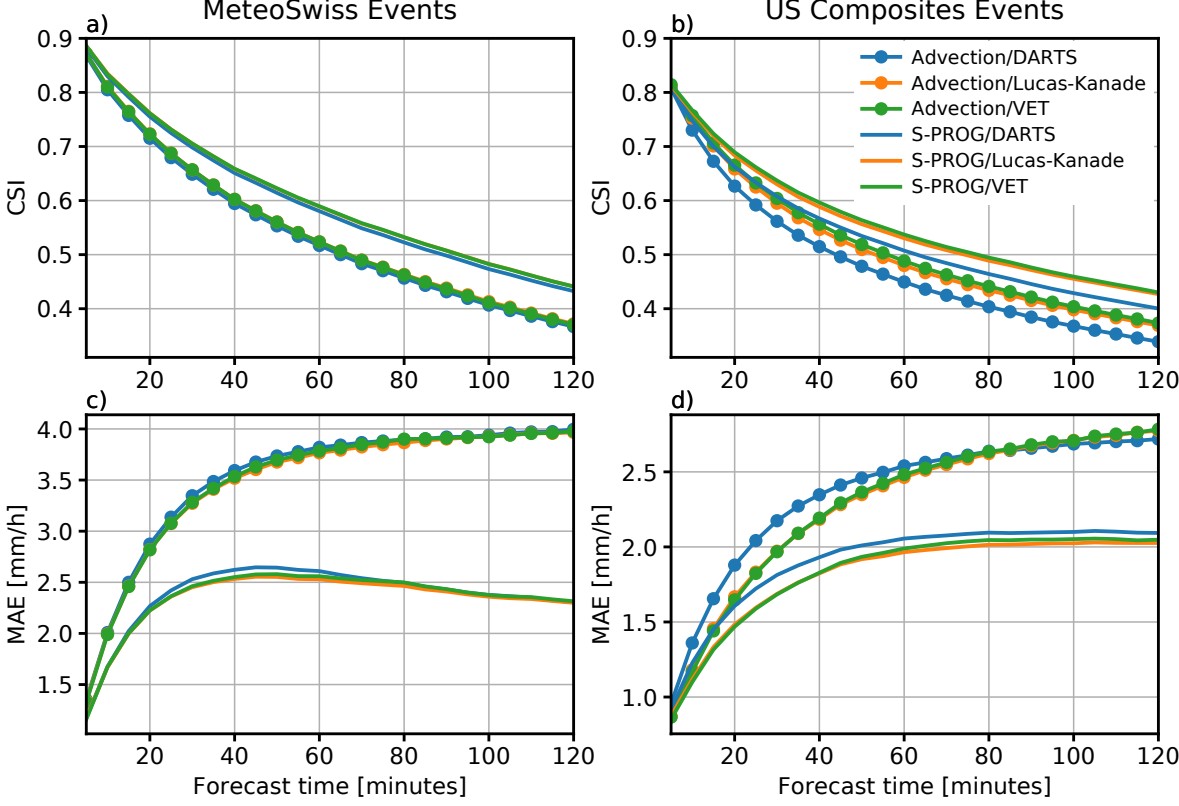

**Figure 11.** Comparison of forecast skill using different optical flow and extrapolation methods. The left panel shows the averaged CSI and MAE for the MeteoSwiss events listed in Table 8, while the right panel shows the same results but for the US events listed in Table 9.

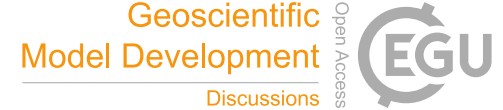

**Figure 12.** Comparison of advection fields obtained by different optical flow methods for a selected precipitation event: US, 2013/04/11 0800 UTC.







**Figure 13.** ROC areas for (a) 0.1 mm h$^{-1}$ and (b) 5 mm h$^{-1}$ thresholds with different ensemble sizes as a function of lead time during the MeteoSwiss events listed in Table 8.





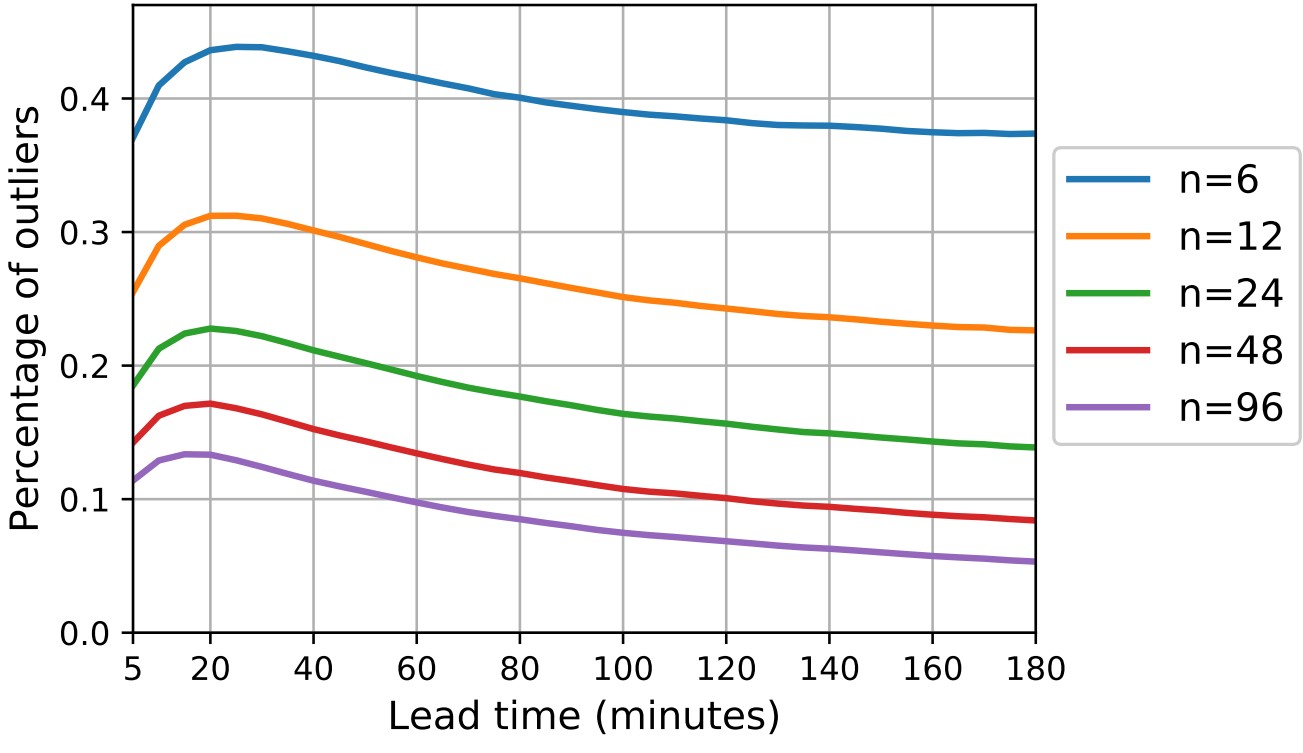

**Figure 14.** Outlier percentages (OP) with different ensemble sizes as a function of lead time during the MeteoSwiss events listed in Table 8.





**Figure 15.** Averaged computation times of pysteps nowcast ensembles with different ensemble sizes and number of parallel threads for the (a) MeteoSwiss and (b) FMI domain. The grid sizes for the domains are 710x640 and 760x1226 pixels, respectively. One-hour nowcasts with 12 time steps of 5 minutes were computed for both domains. The computation times include only the ensemble computation, excluding the optical flow, the initialization of the model and writing the results to disk.





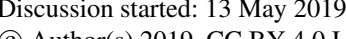

**Figure 16.** Effect of localization in terms of (a) rank histograms and (b) reliability diagrams computed for the 30-minute lead time and 1.0 mm h$^{-1}$ during the MeteoSwiss events (Table 8). The localization window was reduced from the full domain (710 km) to three different local scales (360, 180 and 90 km).



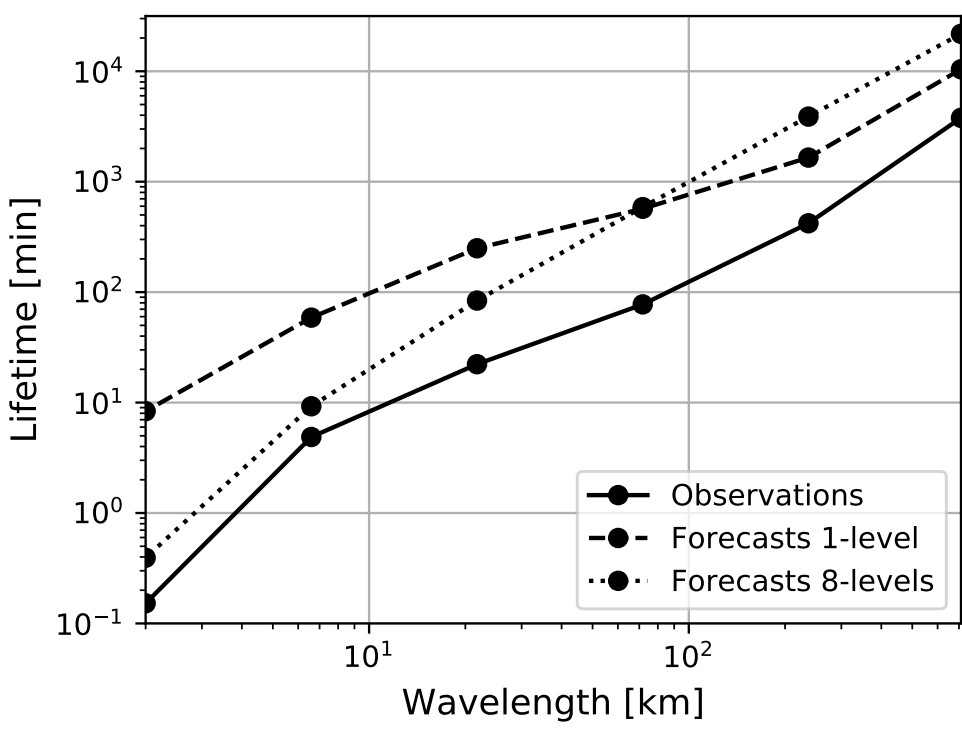

**Figure 17.** Verification of dynamic scaling properties of stochastic nowcasts generated with 1 and 8 cascade levels. All MeteoSwiss events were analyzed, but nowcasts were run only every 4 hours.

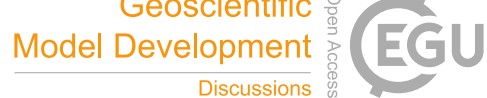

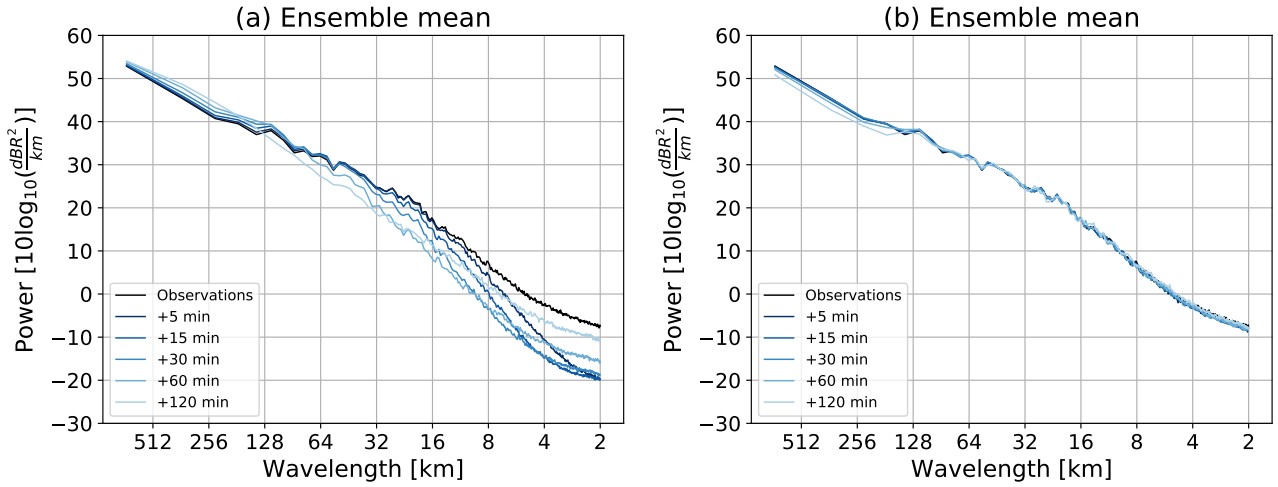

**Figure 18.** Spatial structure analysis of the ensemble mean forecast using a) 8 cascade levels and b) 1 cascade level. Both experiments have an ensemble size of 24 members. MeteoSwiss event #3 was used.





**Figure 19.** Ensemble and probabilistic verification of the cascade experiments for all the MeteoSwiss cases with and without cascade decomposition, and with and without masking. a) Rank histograms at 60 min, b) spread-error relationship, c,d) reliability diagrams at 60 min for probability of rain exceeding 0.1 and 1 mm h$^{-1}$ respectively.





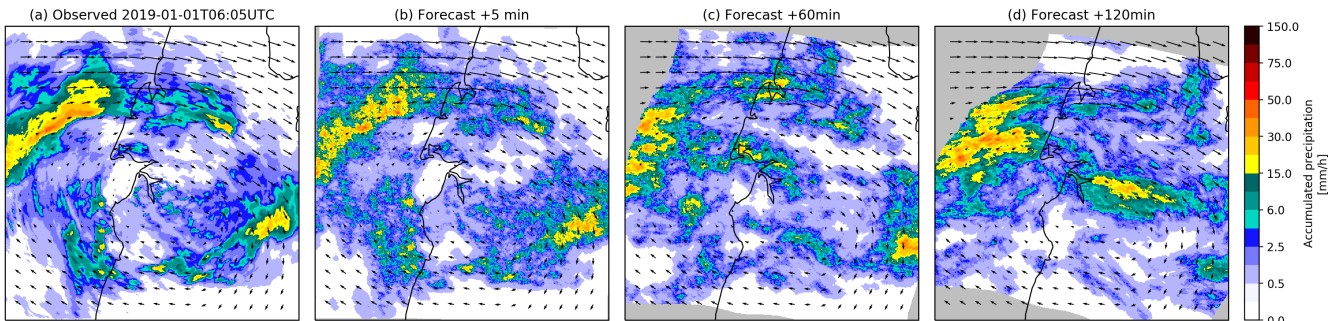

**Figure 20.** Comparison of a member of 5-min rainfall ensemble for (b) +30 minutes, (c) +60 minutes and (d) +120 minutes nowcasts initialized with (a) radar-based rainfall analysis from the Australian radar network valid at time 0605 UTC 01 January 2019 on a 512x512 pixel grid (256x256 km) (event no. 2 in Table 10).



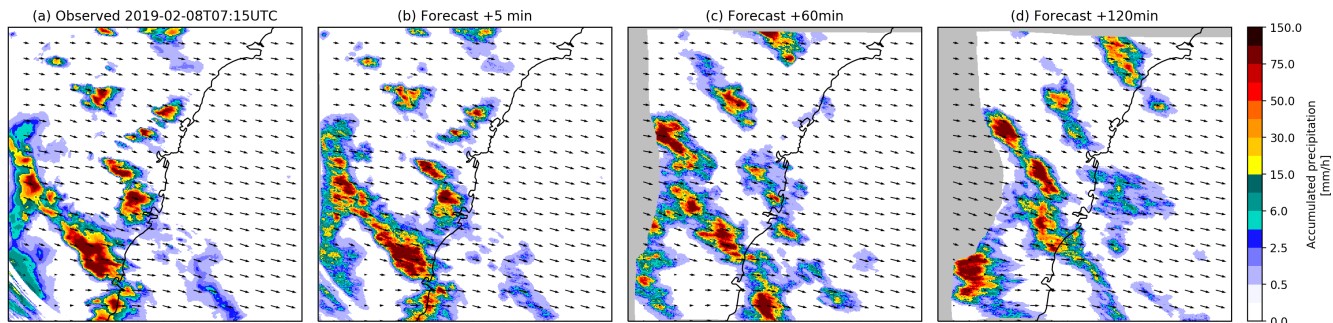

**Figure 21.** Comparison of a member of 5-min rainfall ensemble for (b) +30 minutes, (c) +60 minutes and (d) +120 minutes nowcasts initialized with (a) radar-based rainfall analysis from the Australian radar network valid at time 0715 UTC 08 February 2019 on a 512x512 pixel grid (256x256 km)(event no. 1 in Table 10).





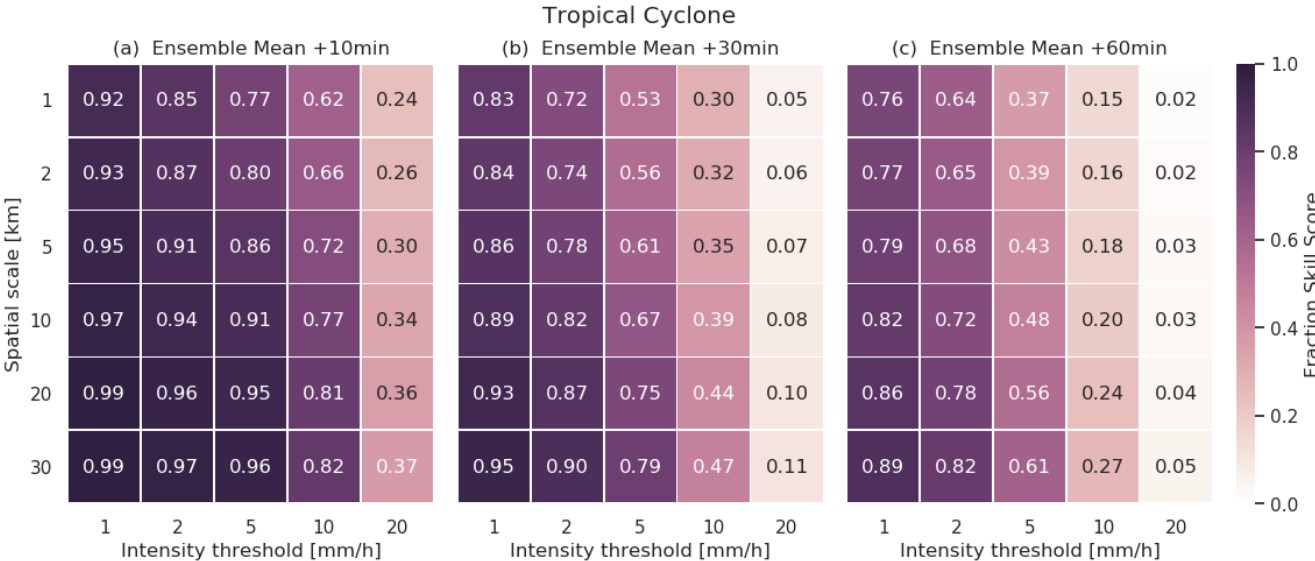

**Figure 22.** Comparison of Fractions Skill Score (FSS) values for (a) +10 minutes, (b) +30 minutes and (c) +60 minutes nowcasts rainfall ensembles for Tropical Cyclone Penny (event no. 1 in Table 10). FSS values were calculated comparing the ensemble mean for each lead time with observations.





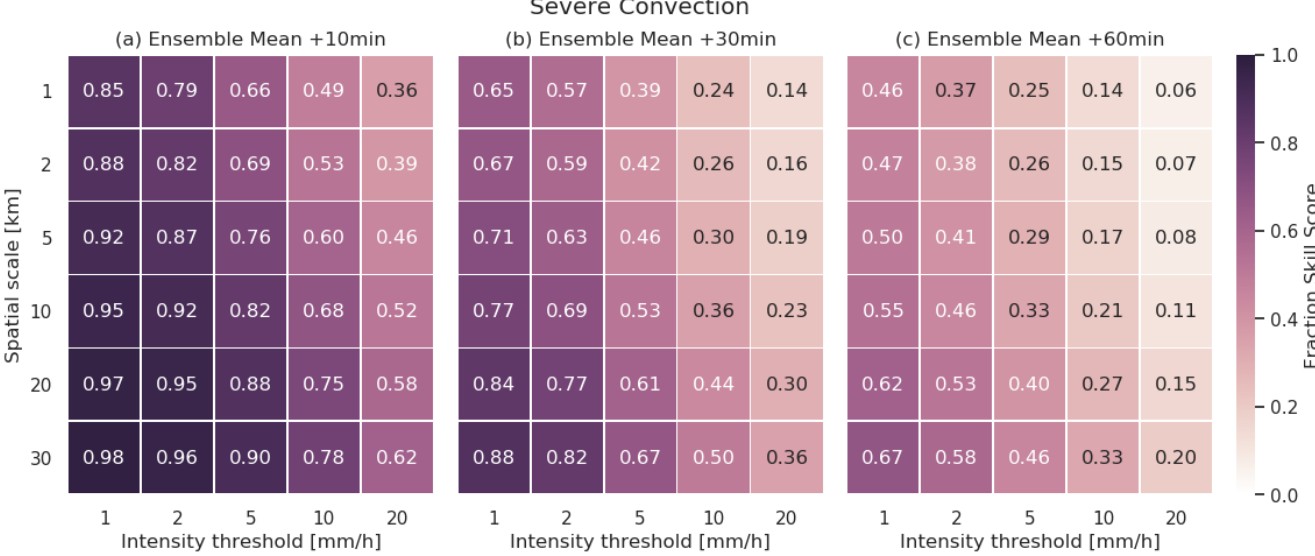

**Figure 23.** Same as Fig. 22 but for the Severe Convection case study (event no. 2 in Table 10).



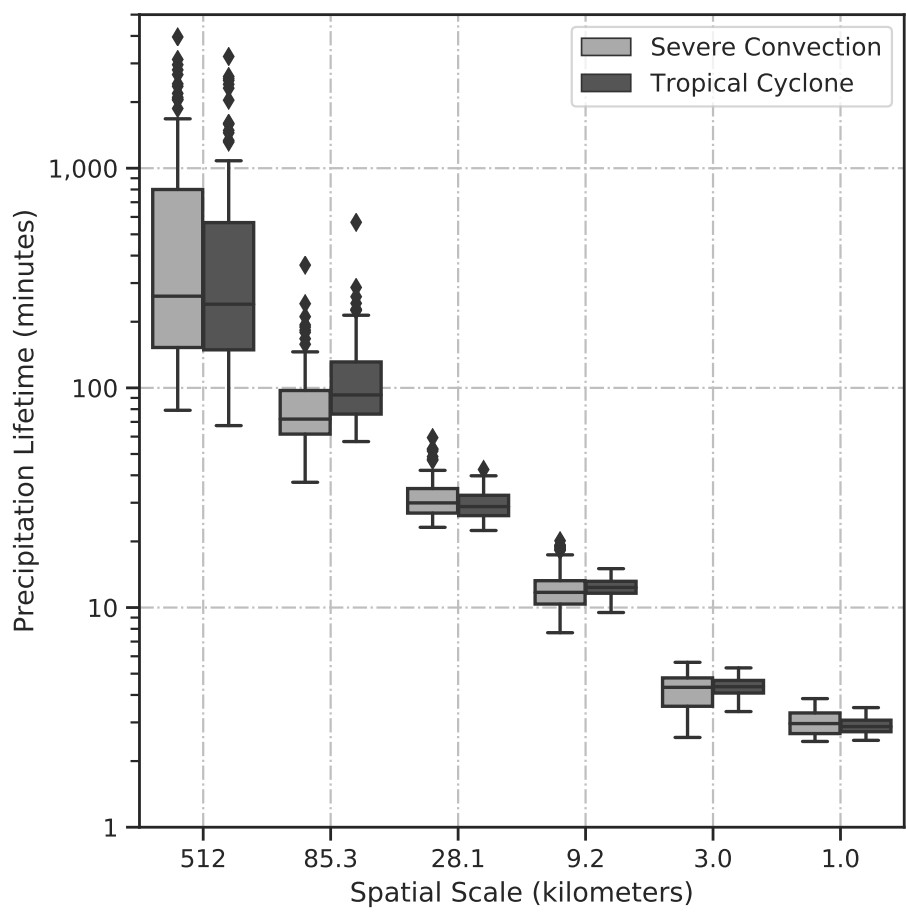

**Figure 24.** Distribution of precipitation lifetime values for each spatial scale during Tropical Cyclone (event no. 1 in Table 10) and Severe Convection (event no. 2 in Table 10) case studies.





**Figure 25.** The deterministic S-PROG nowcast (a), ensemble mean (b) and two ensemble members (c) and (d) of a one-hour STEPS nowcast started at 1600 UTC 28 Sep 2016.





**Figure 26.** Probability of exceeding 0.5 mm/h computed from the STEPS nowcast ensemble shown in Fig. 25 with 24 members.





**Table 1.** Non-exhaustive list (in alphabetical order) of precipitation nowcasting packages that are in principle available to the public. Open-source libraries have their source code available to the general public. Free-licence libraries can be obtained upon request.

| Library | Language | Website | Availability | Reference |
|---|---|---|---|---|
| com-SWIRLS | Python | https://com-swirls.org | free license | Wong et al. (2016) |
| IMPROVER | Python, Shell | https://improver.readthedocs.io | open source | Flowerdew (2018) |
| INCA | C, Fortran, Shell | https://www.zamg.ac.at | free license | Haiden et al. (2011) |
| pysteps | Python | https://pysteps.github.io | open source | this study |
| rainymotion | Python | https://github.com/hydrogo/rainymotion | open source | Ayzel et al. (2018) |
| STEPS | C, C++ | https://www.bom.gov.au (Alan Seed) | free license | Bowler et al. (2006); Seed et al. (2013) |



**Table 2.** External libraries used by pysteps.

| Library | Website | Description |
|---------|---------|-------------|
| h5py | http://www.h5py.org | |
| netCDF4 | http://unidata.github.io/netcdf4-python | input/output |
| PIL | https://github.com/python-pillow/Pillow | |
| OpenCV | http://opencv.org | image processing |
| numpy | http://www.numpy.org | mathematical routines |
| scipy | http://www.scipy.org | |
| FFTW/pyFFTW | http://www.fftw.org | fast Fourier transform |
| | https://github.com/pyFFTW | |
| dask | http://dask.org | parallelization |
| cartopy | https://github.com/SciTools/cartopy | |
| matplotlib | http://matplotlib.org | visualization |
| mpl_toolkits.basemap | http://matplotlib.org/basemap | |



**Table 3.** Overview of pysteps modules.

| Module | Description |
| --- | --- |
| io | reading radar composites and writing nowcast files |
| motion | optical flow methods for motion field computation |
| extrapolation | advection-based extrapolation |
| timeseries | time series methods (e.g. AR models) |
| noise | generation of stochastic noise to perturb precipitation and motion fields |
| cascade | scale-based decomposition of precipitation fields |
| nowcasts | implementation of nowcasting methods |
| postprocessing | statistical post-processing of nowcasts |
| verification | statistical verification of nowcasts and plotting the results |
| visualization | plotting of precipitation and advection fields |
| utils | miscellaneous utility functions (e.g. converting and transforming data and computing the FFT) |



**Table 4.** Overview of the radar QPE composites that have been used to evaluate pysteps. The grid size is given as the number of pixels in the x and y dimensions.

| Dataset | Country | Resolution | Grid size |
|---------|---------|------------|-----------|
| FMI | Finland | 1 km, 5 min | 760x1226 |
| MeteoSwiss | Switzerland | 1 km, 5 min | 710x640 |
| WDSS[1] | United States | 4 km, 5 min | 1361x1056 |
| BoM | Australia | 0.5 km, 5 min | 512x512 |

[1] Upscaled from original data at 1 km resolution (5445x4226)





**Table 5.** The default configuration used in the experiments.

| Parameter | Value |
|---|---|
| optical flow | Lucas-Kanade |
| extrapolation | semi-Lagrangian |
| cascade levels | 8 |
| order of the AR(p) model | 2 |
| precip. intensity perturbations | non-parametric |
| transformation | R to dBR |
| minimum precipitation | $0.1 \text{ mm h}^{-1}$ |
| value for dry pixels | -15 dBR |
| mask method | incremental |
| ensemble size | 24 |
| probability matching | yes |
| seed number | 24 |
| velocity perturbation parameters | $a_{\text{par}} = 2.32$, $b_{\text{par}} = 0.34$, |
| (fitted to pooled FMI | $c_{\text{par}} = -2.65$ |
| and MeteoSwiss data) | $a_{\text{perp}} = 1.91$, $b_{\text{perp}} = 0.34$, |
| | $c_{\text{perp}} = -2.07$ |





**Table 6.** Average computation times of different optical flow methods in the MeteoSwiss and FMI domains (seconds). Domain sizes are given in parentheses.

| 12 cores | | |
|---|---|---|
| **Method** | **MeteoSwiss (710x640)** | **FMI (760x1226)** |
| DARTS | 4.02 | 4.07 |
| Lucas-Kanade | 2.02 | 4.29 |
| VET | 13.73 | 28.98 |
| **1 core** | | |
| **Method** | **MeteoSwiss (710x640)** | **FMI (760x1226)** |
| DARTS | 4.27 | 4.78 |
| Lucas-Kanade | 2.07 | 4.46 |
| VET | 41.65 | 85.14 |



**Table 7.** Precipitation events in Finland (FMI). The duration of each event is 12 hours.

| No. | Date | Start time (UTC) | Description |
|---|---|---|---|
| 1 | 8 Jun 2016 | 13:00 | A low pressure system over Northern Finland causes frontal rain associated with a warm front and frontal rain and convective cells associated with a cold front. The system moves eastward with precipitation areas rotating around its centre. |
| 2 | 15 Jul 2016 | 12:00 | Frontal precipitation approaches measurement area from South. |
| 3 | 28 Sep 2016 | 09:00 | Frontal precipitation intermixed with convection, some scattered convective cells. |
| 4 | 22 Feb 2017 | 22:00 | Wide-spread heavy frontal precipitation associated with a low pressure system traversing over Southern Finland. |
| 5 | 8 Jun 2017 | 04:00 | Narrow and slowly-moving precipitation band. |
| 6 | 14 Jul 2017 | 01:00 | Precipitation starts out as a narrow precipitation band with some scattered convective cells, and later evolves into predominantly convective precipitation. |
| 7 | 4 Aug 2017 | 11:00 | Frontal rain associated with a warm front and some convective activity. |
| 8 | 12 Sep 2017 | 03:00 | Frontal precipitation moves northward and slightly rotates. |
| 9 | 12 Aug 2018 | 05:00 | Frontal precipitation intermixed with convection. Some convective activity, which rotates. Convective activity increases noticeably in a few hours. |
| 10 | 29 Sep 2018 | 16:00 | Frontal precipitation moves eastward and is followed by convective activity. New convective cells are continuously generated at the northern coast of Estonia after the frontal precipitation has passed. |





**Table 8.** Precipitation events in Switzerland (MeteoSwiss). The duration of each event is 12 hours.

| No. | Date | Start time (UTC) | Description |
|---|---|---|---|
| 1 | 16 Apr 2016 | 18:00 | Prefrontal precipitation induced by a low pressure system over the North Sea. Lines of convection develop over western Switzerland. |
| 2 | 11 Jul 2016 | 13:00 | An approaching cold front causes widespread convective activity in a south-westerly flow. |
| 3 | 31 Jan 2017 | 10:00 | A strong north-westerly flow causes orographic blocking on the northern slopes of the Alps resulting in widespread precipitation. |
| 4 | 14 Jun 2017 | 13:00 | Fairly uniform pressure distribution across Central Europe, scattered convection develops during the afternoon. |
| 5 | 24 Jun 2017 | 22:00 | Prefrontal activity with intense thunderstorms south of the Alps. Measured peak intensity reached 33.5 mm in 10 min and presence of large size hail stones (3-5 cm) was observed. |
| 6 | 27 Jun 2017 | 20:00 | A frontal passage during the night induces organized convection over the domain and important prefrontal convective activity on the southern side of the Alps. |
| 7 | 19 Jul 2017 | 13:00 | In a south-westerly flow, development of large convective cells over central Switzerland. |
| 8 | 21 Jul 2017 | 13:00 | Flat pressure distribution across central Europe, South-westerly flow associated to a low over the British Islands. Clusters of intense thunderstorms develop over western Switzerland. |
| 9 | 29 Jul 2017 | 13:00 | South-westerly flow connected to large depression over British Islands. Large clusters of convection develop south of the Alps. |
| 10 | 31 Aug 2017 | 14:00 | Strong south-westerly flow over the Alps associated to a cold front. Important lines of stationary convection affect the southern Alps, while more stratiform precipitation occurs in the West and North of Switzerland. |





**Table 9.** Precipitation events from USA. The duration of each event is 12 hours.

| No. | Date | Start time (UTC) | Description |
|---|---|---|---|
| 1 | 16 Apr 2011 | 08:00 | Large extratropical cyclone. The low pressure center was located over the Great Lakes with a strong cold front extending south. |
| 2 | 15 Nov 2011 | 23:00 | Frontal precipitation associated with a stationary front in southeastern US. |
| 3 | 04 Apr 2013 | 18:00 | Two widespread precipitation systems produced by two cyclonic systems over the US, located in the north-west and south-east of the US. |
| 4 | 11 Apr 2013 | 00:00 | Mid-latitude cyclone over the eastern US with the eastern line of precipitation caused by a cold front extending in the south-north direction from eastern Texas to central Missouri, and in the west-east direction from Missouri to the south of the New York state. |
| 5 | 18 May 2013 | 06:00 | Mesoscale Convective Systems (MCSs) located in northern and south-eastern US. |
| 6 | 27 May 2017 | 00:00 | MCSs developed over central and north-western US, along with a cyclonic precipitation system located in the south-eastern Canada. |





**Table 10.** Precipitation events from Australia (BoM). The duration of each event is 12 hours.

| No. | Date | Start time (UTC) | Description |
|-----|------|------------------|-------------|
| 1 | 01 Jan 2019 | 00:00 | Tropical Cyclone Penny moving from the Gulf of Carpenteria and making landfall on the western Cape York Peninsula coastline just south of Weipa C-band Doppler radar. |
| 2 | 08 Feb 2019 | 05:00 | Severe convection activity observed by the S-band polarimetric radar of Terry Hills near Sydney. Convective cells are continuously generated inland New South Wales and intensifying as they move east. This event produced thunderstorms, heavy rainfall and flash flooding in the City of Parramatta and Western Sydney suburbs. |