# Peer review of "Pysteps: an open-source Python library for probabilistic precipitation nowcasting (v1.0)"

_Geoscientific Model Development, 2019_

## Referee Comment (RC1) · Wang Chun Woo (Referee) · 11 Jun 2019

Peer Review Comments on Pysteps

This paper is well written, clearly and comprehensively describing a novel initiative to open-source an operational-ready nowcasting software package. It explains the science and techniques behind included algorithms, gives some examples, provides verification results and concludes with a vision to further the collaboration on nowcasting techniques and system.

This paper should be accepted for publications, after some modifications to address the following issues:

[Figure]

**1 General Comments**

1. The determination and adoption of reflectivity – rainfall relationships are one of the most critical challenges in QPE and QPF, especially for tropical and sub-tropical regions. Perhaps the authors could include a section to mention what functions pysteps provide in this regard. For example, whether it supports adoption of user-defined a  b values in $Z = aR^b$, whether it makes use of dual-pol data for QPE, whether it can derive a  b from a set of climatological data etc.

**2 Specific Comments**

1. Page 1 Line 7: It is mentioned that pysteps library supports "standard input/output file formats". But radar and rain gauge data come in a high variety of formats. Perhaps it would be more appropriate to write "various input/output file formats". Also suggests listing out the supported data formats in a table.

2. Page 1 Line 15: My understanding is that the definition of nowcasting does not confine itself to "by extrapolation". For longer range nowcast (3 hours and beyond), the use of NWP is indeed more common. Recently, there also emerged other methods, such as deep learning nowcast as elaborated in the following papers: https://papers.nips.cc/paper/5955-convolutional-lstm-network-a-machine-learning-approach-for-precipitation-nowcasting.pdf, https://arxiv.org/abs/1706.03458

3. Page 1 Line 15: Suggest citing the WMO Guideline on Nowcasting: https://library.wmo.int/doc_num.php?explnum_id=3795

4. Page 8 Line 14: Suggests mentioning that some examples of parameters are given in Table 5, if indeed so.

5. Page 9 Line 7: Also commonly known as "frequency matching" in operational meteorology.

6. Page 10 Line 4: Suggests citing Cartopy as requested (https://scitools.org.uk/cartopy/docs/latest/citation.html)

7. Page 10 Line 10: It would also be beneficial to operational meteorology for comparing performance of various nowcast algorithms, and for running multi-model ensemble nowcast.

8. Page 13 Line 7: May consider including the contingency table and the formulas of the performance measures, due to historical confusion between FAR, POFD and "False Alarm Rate"

9. Page 19 Line 6: While noting that the 1 level has over-dispersion in the bin range 13-22, the 23rd bin of 1 level + mask looks better than the 8 level + mask. Would the over-dispersion of 8 level + mask at bin 23 be improved by having more members?

10. Page 22 Line 16: Other potential enhancements include i) use of satellite nowcast parameters based on EUMETSAT NWCSAF algorithms and products (e.g. CI, RDT) to predict the growth of storm cells; ii) use of dual-polarization radar data to enhance QPE and to detect hails, gust etc.

11. Page 22 Line 21: Suggest also including SWIRLS Ensemble Rainstorm Nowcast (SERN) based on ROVER (https://doi.org/10.3390/atmos8030048)

12. Page 22 Line 25: An obvious and crucial application of nowcasting system is to support the operations of rainstorm, thunderstorm and severe weather warnings. Suggest mentioning this also.

13. Page 37 Figure 5: For the reliability diagram on the right, suggests also plotting out the data points used to define the curve if not too dense

14. Page 37 Figure 6: For the reliability diagram on the right, suggests also plotting out the data points used to define the curve if not too dense

**3 Technical Corrections**

1. Page 1 Line 14: "world meteorological organization" should read "World Meteorological Organization (WMO)"

2. Page 2 Line 32: "com-SWIRLS" should read "Com-SWIRLS"

3. Page 3 Line 1: "... by the Hong Kong Observatory" should read "... by the Regional Specialized Meteorological Centre (RSMC) for Nowcasting operated by the Hong Kong Observatory (HKO)".

4. Page 6 Line 3: Missing second term in equation (5)?

5. Page 13 Line 12: "For a reliable forecast" should read "For a perfectly reliable forecast"

6. Page 59 Table 1 Entry 1: Com-SWIRLS employs C++ for several time-critical modules, in addition to Python.

7. Page 59 Table 1 Entry 1: Com-SWIRLS is free and open source for all National Meteorological and Hydrological Services (NMHS) under WMO. I understand that the authors are trying to divide the software into two groups, i.e. "free license" vs "open source". Perhaps a footnote for Com-SWIRLS would make it clearer.

---

## Referee Comment (RC2) · Anonymous Referee #2 · 21 Jun 2019

The manuscript deals with precipitation nowcasting and describes a package of programs designed for practical predictions as well as further investigations of forecasting techniques. The article consists of two main parts. In the first part, a brief description of the used methods is given, while in the second part, procedures that are ready for use are described. Basic verification and sensitivity analysis of some parameters of the applied methods are performed and results are shown. The article is very comprehensive and worth of publishing, however, I have several comments concerning the content that are listed below. The article and especially its introduction is written in a too optimistic way that nowcating can solve the prediction of severe weather and in this context, it is mentioned that current NWP models are not able to predict phenomena on convective scale. In the introduction, a lead time of 6h for nowcasting is mentioned.

[Figure]

However, in my opinion and based on my experience, the reality is different and reliability of precipitation forecast depends on the type of predicted precipitation mainly. Cases presented in the paper, if I can judge, are characterized by large rainfall areas, which, as a rule, can be sufficiently accurately predicted by extrapolation-based methods for several hours in advance. For this type of situations, however, NWP model predictions give also good results. Conversely, cases with isolated convective storms that according to me are not treated in the paper are very difficult to forecast by extrapolation methods reasonably, however, they may cause very dangerous local flash floods. Moreover, I find it a pity that the authors did not try to verify the proposed methods for a continuous series of data, e.g. covering 3 months. I wonder if the proposed methods would produce as good results in such a case as they are presented in the paper. Some publications have indicated (e.g. Mejsnar et al., 2018. Limits of precipitation nowcasting by extrapolation of radar reflectivity for warm season in Central Europe. Atmos. Res., 213, 288-301) that extrapolation of convective precipitation and also NWP model forecast is very difficult in inland areas. Although I do not require performing additional tests or verifications, I still consider it fair to mention the known prediction problems in both the introduction and the conclusion sections.

Besides, the applied technique based on application of FFT needs further additional comments. Is this technique reasonable in case of isolated convection, when large majority of the area evinces no precipitation? Section 5.3 Line 29 Could you briefly describe what you mean by "localization"? Section 5.4 Line 10 You write: "Thus, our main hypothesis is that dynamic scaling properties are necessary to produce ensembles with realistic temporal evolution and dispersion of precipitation across spatial scales." I am not sure whether I understand what you mean. Could you kindly add some comments on "realistic temporal evolution" and its consequences? Looking at animations (line 27), I agree that the cascade decomposition smooths and decreases precipitation. It seems to me that the model expects and forecasts dissipating of storms. However, this is realistic only under several specific conditions. Under other conditions, increase of precipitation can occur. In any case, the presented smoothed fields in the animation
do not look very realistic to me and the fact that they provide better RMSE verification values is a simple result of the known general feature of RMSE. To sum up, I find the article and the software very useful but readers and especially users should be aware that any forecasting technique has also its weak points, at least at present.

---

## Author Comment (AC1) · 6 Aug 2019

**Responses to reviewers for manuscript**

**Pysteps: an open-source Python library for probabilistic precipitation nowcasting (v1.0)**

- Reviewers comments in black
- Responses to reviewers in blue

**Reviewer #1: Wang Chun Woo**

This paper is well written, clearly and comprehensively describing a novel initiative to open-source an operational-ready nowcasting software package. It explains the science and techniques behind included algorithms, gives some examples, provides verification results and concludes with a vision to further the collaboration on nowcasting techniques and system. This paper should be accepted for publication, after some modifications to address the following issues:

We thank Wang-Chun Woo (Reviewer #1) for the precise comments and suggestions. Our point-by-point answers are listed hereafter.

**1. General Comments**

1. The determination and adoption of reflectivity – rainfall relationships are one of the most critical challenges in QPE and QPF, especially for tropical and subtropical regions. Perhaps the authors could include a section to mention what functions pysteps provide in this regard. For example, whether it supports adoption of user-defined **a b** values in $Z=aR^b$, whether it makes use of dual-pol data for QPE, whether it can derive **a** and **b** from a set of climatological data, etc.

   Because the focus is on QPF rather than QPE, pysteps does not include methods to derive Z-R relationships. The nowcasting methods take rain rate arrays as inputs, and generating the QPE product is beyond the scope of pysteps. Currently only the basic conversion from Z to R is implemented. This is because many meteorological institutes store radar mosaics in reflectivity (dBZ) for historical reasons.

   The user can extend pysteps to include different input formats and custom conversions to rain rates. Currently the io.importers module implements reading radar mosaics from FMI, MeteoSwiss, BoM, KNMI and OPERA. The user can implement custom importers for reading their input data and do the conversion to rain rate. The a and b parameters of the Z-R relationship can be included as attributes (metadata) in the importer. Alternatively, the conversion can be done by using the to_rainrate function in the utils.conversion module. This function supports user-defined a and b parameters.

The first paragraph of Section 3.5 has been revised to make a more clear statement of the input data, and also to add some of the points mentioned above. The corresponding part of Figure 4 was also revised.

**2. Specific Comments**

1. Page 1 Line 7: It is mentioned that pysteps library supports "standard input/output file formats". But radar and rain gauge data come in a high variety of formats. Perhaps it would be more appropriate to write "various input/output file formats".

   We replaced "standard" by "various" as suggested.

   I also suggest listing out the supported data formats in a table.

   Since the supported in/out data formats are technical details that are rapidly evolving, in Section 3.2, we refer the reader to the pysteps official documentation for more details.

2. Page 1 Line 15: My understanding is that the definition of nowcasting does not confine itself to "by extrapolation". For longer range nowcast (3 hours and beyond), the use of NWP is indeed more common. Recently, there also emerged other methods, such as deep learning nowcast as elaborated in the following papers:
   https://papers.nips.cc/paper/5955-convolutional-lstm-network-a-machine-learning-approach-for-precipitation-nowcasting.pdf,https://arxiv.org/abs/1706.034583.

   Indeed, the term nowcasting, strictly speaking, refers solely to a forecasting range, rather to a particular forecasting method. The first paragraph of the introduction has been changed so that in addition to extrapolation, statistical and NWP models are also mentioned. In addition, reference to the above paper was added to the first paragraph of Section 1.1.

3. Page 1 Line 15: Suggest citing the WMO Guideline on Nowcasting:
   https://library.wmo.int/doc_num.php?explnum_id=37954.

4. Page 8 Line 14: Suggests mentioning that some examples of parameters are given in Table 5, if indeed so.

   We added a reference to Table 5.

5. Page 9 Line 7: Also commonly known as "frequency matching" in operational meteorology.
   We would like to point out that precisely speaking the technique described in Section 2.8 is not frequency matching. This is because the CDFs are matched, not frequencies. In addition, we slightly revised the paragraph. The last sentence was removed, and now we cite to Foresti et al. 2016, where a similar technique was used.

6. Page10 Line 4: Suggest citing Cartopy as requested
   (https://scitools.org.uk/cartopy/docs/latest/citation.html)

   We have provided, when available, references to all packages listed in Table 2.

7. Page 10 Line 10: It would also be beneficial to operational meteorology for comparing the performance of various nowcast algorithms, and for running multi-model ensemble nowcast.

   Thanks for the suggestion. We added the possible uses in operational meteorology.

8. Page 13 Line 7: May consider including the contingency table and the formulas of the performance measures, due to historical confusion between FAR, POFD and "False Alarm Rate"

   In that paragraph we already mentioned the possible confusion. Nonetheless, we added a reference with more details on the contingency tables and the formulas of the categorical scores used (page 13, lines 12-14).

9. Page 19 Line 6: While noting that the 1 level has over-dispersion in the bin range 13-22, the 23rd bin of 1 level + mask looks better than the 8 level + mask. Would

the over-dispersion of 8 level + mask at bin 23 be improved by having more members?

The experiment with 8 levels and the precipitation mask in fact produces an underdispersive ensemble (notice the last bin is larger than the reference, meaning that there are too many observations falling above the ensemble range).
By looking at Fig. 14, we can see that increasing the ensemble size decreases the underdispersion. Hence, yes, a larger ensemble size improves the dispersion of the ensemble. We added a sentence about this to the corresponding paragraph in the manuscript.

10. Page 22 Line 16: Other potential enhancements include i) use of satellite nowcast parameters based on EUMETSAT NWCSAF algorithms and products (e.g. CI,RDT) to predict the growth of storm cells; ii) use of dual-polarization radar data to enhance QPE and to detect hail, gusts etc.

These suggestions are excellent examples of exciting research topics in which pysteps may prove useful as a computational tool and a development framework. However, i) represents an advanced application that arguably will be difficult to implement in pysteps in the short or medium term. For ii), we point out that it is not actually an enhancement of pysteps, but the input data. Referring to our response to General Comment 1., this is beyond the scope of pysteps.

11. Page 22 Line 21: Suggest also including SWIRLS Ensemble Rainstorm Nowcast (SERN) based on ROVER (https://doi.org/10.3390/atmos8030048).

We added SERN to the bibliography with the corresponding reference.

12. Page 22 Line 25: An obvious and crucial application of nowcasting system is to support the operations of rainstorms, thunderstorms and severe weather warnings. Suggest mentioning this also.

Those applications can be considered to be part of the field of nowcasting, while that particular paragraph discusses the applications beyond the nowcasting.

Nonetheless, we modified the introduction section to explicitly mention the support of severe weather warning operations (Page 2 lines 1-2).

13. Page 37 Figure 5: For the reliability diagram on the right, suggests also plotting out the data points used to define the curve if not too dense.

Thank you for the suggestion. Unfortunately the number of data points (i.e., thresholds) is indeed too high to individually plot each one of them.

14. Page 37 Figure 6: For the reliability diagram on the right, suggests also plotting out the data points used to define the curve if not too dense.

See answer to comment 13.

**3. Technical Corrections**

1. Page 1 Line 14: "world meteorological organization" should read "World Meteorological Organization (WMO)".
Change made.

2. Page 2 Line 32: "com-SWIRLS" should read "Com-SWIRLS"\
Change made

3. Page 3 Line 1: ". . .by the Hong Kong Observatory" should read ". . .by the Regional Specialized Meteorological Centre (RSMC) for Nowcasting operated by the Hong Kong Observatory (HKO)".
Change made.

4. Page 6 Line 3: Missing second term in equation (5)?
We corrected the formatting problem (removed the break line).

5. Page 13 Line 12: "For a reliable forecast" should read "For a perfectly reliable forecast".
Change made.

6. Page 59 Table 1 Entry 1: Com-SWIRLS employs C++ for several time-critical modules, in addition to Python.
Change made.

7. Page 59 Table 1 Entry 1: Com-SWIRLS is free and open source for all National Meteorological and Hydrological Services (NMHS) under WMO. I understand that

the authors are trying to divide the software into two groups, i.e. "free license" vs "open source". Perhaps a footnote for Com-SWIRLS would make it clearer.
As suggested, we now indicate Com-SWIRLS as open source and specify in a footnote that this applies to NMHS only.

**Reviewer #2:**

The manuscript deals with precipitation nowcasting and describes a package of programs designed for practical predictions as well as further investigations of forecasting techniques. The article consists of two main parts. In the first part, a brief description of the used methods is given, while in the second part, procedures that are ready for use are described. Basic verification and sensitivity analysis of some parameters of the applied methods are performed and results are shown. The article is very comprehensive and worth of publishing, however, I have several comments concerning the content that are listed below.

We would like to thank reviewer #2 for the comments and suggestions. We have decided to split review #2 into a set of individual points and provided our responses accordingly.

**Comments:**

1. The article and especially its introduction is written in a too optimistic way that nowcasting can solve the prediction of severe weather and in this context, it is mentioned that current NWP models are not able to predict phenomena on convective scale.

   We updated the introduction to describe the advantages of extrapolation-based nowcasting vs NWP, replacing the discussion about the NWP accuracy.

   "Weather radars are ideally suited for providing the input data for extrapolation-based precipitation nowcasting at high resolution, namely spatial scales under 2 km and time ranges between 5 minutes and 3 hours (Berne et al., 2004). Despite recent advances in numerical weather prediction (NWP, e.g. Sun et al., 2014), extrapolation-based nowcasting remains the primary approach at such space and time scales, typically outperforming NWP forecasts in the first 2-5 hours, depending on the weather situation, domain and NWP model characteristics (e.g. Berenguer et al., 2012; Mandapaka et al., 2012; Simonin et al., 2017; Jacques et al., 2018)." **Page 2, lines 6-11.**

2.  In the introduction, a lead time of 6h for nowcasting is mentioned. However, in my opinion and based on my experience, the reality is different and reliability of precipitation forecast depends on the type of predicted precipitation mainly. Cases presented in the paper, if I can judge, are characterized by large rainfall areas, which, as a rule, can be sufficiently accurately predicted by extrapolation-based methods for several hours in advance. For this type of situations, however, NWP model predictions give also good results. Conversely, cases with isolated convective storms that according to me are not treated in the paper are very difficult to forecast by extrapolation methods reasonably, however, they may cause very dangerous local flash floods.

    Nowcasting is commonly defined as short-term forecast, generally for the next few hours. As such, the definition is independent of the meteorological situation or the forecasting method. For example, the world meteorological organization defines nowcasting as  "forecasting with local detail, by any method, over a period from the present to 6 hours ahead, including a detailed description of the present weather" (https://library.wmo.int/doc_num.php?explnum_id=3795, last access 2019/08/05).

    Nevertheless, we are aware that the type of precipitation and its predictability will have an influence on the quality of the forecast. The scope of the manuscript is limited to the description of an open-source library that provides access to state-of-the-art nowcasting methods and additional tools to facilitate the analysis and presentation of the results. A detailed analysis of the limitations of different nowcasting methods under different meteorological situations, or a side-by-side comparison between extrapolation and NWP forecasts, is indeed interesting but outside the scope of the paper.

    Finally, we would like to make the reviewer aware that our datasets also contain a quite important fraction of summer convective events, including isolated convection (see e.g. Table 8 for Switzerland).

3.  Moreover, I find it a pity that the authors did not try to verify the proposed methods for a continuous series of data, e.g. covering 3 months. I wonder if the proposed methods would produce as good results in such a case as they are presented in the paper. Some publications have indicated (e.g. Mejsnar et al., 2018. Limits of precipitation nowcasting by extrapolation of radar reflectivity for

warm season in Central Europe. Atmos. Res., 213, 288-301) that extrapolation of convective precipitation and also NWP model forecast is very difficult in inland areas. Although I do not require per forming additional tests or verifications, I still consider it fair to mention the known prediction problems in both the introduction and conclusions sections.

As mentioned above, we aim at  describing an open-source and community-driven Python library, pysteps, that provides access to state-of-the-art nowcasting methods and additional tools to facilitate the analysis and presentation of the results.

For clarity, we updated the last part of the introduction to make the scope more clear:

"In this article, we present pysteps, an open-source and community-driven Python library for probabilistic precipitation nowcasting." Page 3, Lines 8-9.

The number of events included in the study might seem somewhat limited at first, but 10 precipitation events of 12 hours each represent for a given region a significant amount of precipitation compared to what could be expected over a randomly selected 3-month period.

One should also consider that we provided results for four contrasting regions (Finland, Switzerland, continental United States and Australia), which arguably represent a wide range of climates and domain characteristics and therefore offer a robust assessment of the pysteps library.

4. Besides, the applied technique based on application of FFT needs further additional comments. Is this technique reasonable in case of isolated convection, when a large majority of the area evinces no precipitation?

The FFT approach is used in a number of well-known ensemble precipitation nowcasting techniques that are both found in the literature (e.g., Bowler et al., 2006; Metta et al., 2009; Berenguer et al., 2011; Atencia and Zawadzki, 2014) and used operationally (e.g., Foresti et al., 2016).
Our work, therefore, does not introduce any novelty or more scientific insight in this sense, but merely provides access to state-of-the-art nowcasting methods, thus enabling the investigation of scientific questions. In this sense, the limits of the FFT approach in case of isolated convection would certainly represent an

interesting topic for future research in which pysteps could serve as a computational tool.

5. Section 5.3 Line 29 Could you briefly describe what you mean by "localization"?

   We revised the first paragraph of Section 5.3 to improve the presentation and to make the main point more clear. That is, the nowcasting model is applied in small subdomains instead of the whole domain (which assumes spatial homogeneity of the precipitation field).

6. Section 5.4 Line 10 You write: "Thus, our main hypothesis is that dynamic scaling properties are necessary to produce ensembles with realistic temporal evolution and dispersion of precipitation across spatial scales." I am not sure whether I understand what you mean.

   "Dynamic scaling" refers to a well-known relationship between spatial and temporal structures of rainfall (e.g., Venugopal et al., 1999). It represents the scale-dependence of the predictability of precipitation, that is, the observation that large scale structures are more predictable than small scale structures (e.g., convective cells). Dynamic scaling properties can be observed as a power-law relating the rate of temporal evolution (i.e., lifetime) to the spatial scale, as explained in Venugopal et al. (1999). In addition, the amount of perturbations added to the forecast field is inversely related to predictability via equation (6). We agree that this link between dynamic scaling and the ensemble dispersion was not well-explained in Section 5.4, line 10. Therefore, we rephrased the sentence as follows:

   "Thus, our main hypothesis is that dynamic scaling properties are necessary to produce a realistic temporal evolution (lifetime) of precipitation across spatial scales. Consequently, this would give correct ensemble dispersion because the standard deviation of the perturbations is inversely related to predictability via equation (6)".

7. Could you kindly add some comments on "realistic temporal evolution" and its consequences? Looking at animations (line 27), I agree that the cascade decomposition smooths and decreases precipitation. It seems to me that the model expects and forecasts dissipating of storms. However, this is realistic only under several specific conditions. Under other conditions, increase of precipitation can occur. In any case, the presented smoothed fields in the

animation do not look very realistic to me and the fact that they provide better RMSE verification values is a simple result of the known general feature of RMSE.

In the pysteps GitHub repo, we provided example animations (*.gif files) of different types of forecasts configurations (https://github.com/pySTEPS/pysteps-publication/tree/master/animations). Indeed, smoother forecasts, for example the ensemble mean (*_mean_*.gif), generally lead to lower RMSE. With "realistic temporal evolution" we referred to the one of the stochastic ensemble members (*_stoch_*.gif). The ensemble mean, which can also be produced using S-PROG, is not a realization of the future atmospheric state. It is merely a statistical summary, as could be the median or a quantile (e.g. 90%).

To help the reader, we have specified that the realistic evolution refers to the ensemble members and not to the ensemble mean, which is of course smoother.

Finally, without external information (e.g. satellite or NWP) or a detailed analysis of the impact of orography on the growth and decay of precipitation cells, the stochastic model cannot predict systematic growth and decay trends. It only produces precipitation fields with a realistic variability in space and time that adequately represent the forecast uncertainty.

In case of the animation of one stochastic member (201609281600_stoch_8levels.gif), one should consider that this represents a single random realization and cannot provide general conclusions on growth and dissipation of individual storms, but merely provide one possible scenario.

8. To sum up, I find the article and the software very useful but readers and especially users should be aware that any forecasting technique has also its weak points, at least at present.

As suggested, we tried to better specify the limitations of extrapolation-based nowcasting. See our responses to the above comments.

**References**

- Atencia, A., and I. Zawadzki, 2014: A comparison of two techniques for generating nowcasting ensembles. Part I: Lagrangian ensemble technique. *Mon. Weather Rev.*, **142 (11)**, 4036–4052.
- Berenguer, M., D. Sempere-Torres, and G. G. Pegram, 2011: SBMcast An ensemble nowcasting technique to assess the uncertainty in rainfall forecasts by Lagrangian extrapolation. *J. Hydrol.*, **404 (3)**, 226–240.
- Berenguer, Marc, Madalina Surcel, Isztar Zawadzki, Ming Xue, and Fanyou Kong. 2012. "The Diurnal Cycle of Precipitation from Continental Radar Mosaics and Numerical Weather Prediction Models. Part II: Intercomparison among Numerical Models and with Nowcasting." *Monthly Weather Review* 140 (8): 2689–2705. https://doi.org/10.1175/mwr-d-11-00181.1.
- Bowler, N. E., C. E. Pierce, and A. W. Seed, 2006: STEPS: A probabilistic precipitation forecasting scheme which merges an extrapolation nowcast with downscaled NWP. *Q. J. R. Meteorol. Soc.*, **132 (620)**, 2127–2155.
- Foresti, L., M. Reyniers, A. Seed, and L. Delobbe, 2016: Development and verification of a real-time stochastic precipitation nowcasting system for urban hydrology inBelgium.*Hydrol. Earth Syst. Sci.*,**20 (1)**, 505–527.
- Mandapaka, Pradeep V, Urs Germann, Luca Panziera, and Alessandro Hering. 2012. "Can Lagrangian Extrapolation of Radar Fields Be Used for Precipitation Nowcasting over Complex Alpine Orography?" *Weather and Forecasting* 27 (1): 28–49. https://doi.org/10.1175/WAF-D-11-00050.1.
- Metta, S., J. von Hardenberg, L. Ferraris, N. Rebora, and A. Provenzale, 2009: Precipitation nowcasting by a spectral-based nonlinear stochastic model. *J. Hydrometeorol.*, **10 (5)**, 1285–1297.
- Venugopal, V., E. Foufoula-Georgiou, and V. Sapozhnikov, 1999: Evidence of dynamic scaling in space-time rainfall. *J. Geophys. Res. Atmos.*, **104 (D24)**, 31 599–31 610.